# AdvAD: Exploring Non-Parametric Diffusion for Imperceptible Adversarial Attacks

**Jin Li**[1], **Ziqiang He**[1], **Anwei Luo**[1], **Jian-Fang Hu**[1], **Z. Jane Wang**[2], **Xiangui Kang**[1]*

[1]Guangdong Key Lab of Information Security,
School of Computer Science and Engineering, Sun Yat-Sen University
[2]Electrical and Computer Engineering Dept, University of British Columbia

## Abstract

Imperceptible adversarial attacks aim to fool DNNs by adding imperceptible perturbation to the input data. Previous methods typically improve the imperceptibility of attacks by integrating common attack paradigms with specifically designed perception-based losses or the capabilities of generative models. In this paper, we propose Adversarial Attacks in Diffusion (AdvAD), a novel modeling framework distinct from existing attack paradigms. AdvAD innovatively conceptualizes attacking as a non-parametric diffusion process by theoretically exploring basic modeling approach rather than using the denoising or generation abilities of regular diffusion models requiring neural networks. At each step, much subtler yet effective adversarial guidance is crafted using only the attacked model without any additional network, which gradually leads the end of diffusion process from the original image to a desired imperceptible adversarial example. Grounded in a solid theoretical foundation of the proposed non-parametric diffusion process, AdvAD achieves high attack efficacy and imperceptibility with intrinsically lower overall perturbation strength. Additionally, an enhanced version AdvAD-X is proposed to evaluate the extreme of our novel framework under an ideal scenario. Extensive experiments demonstrate the effectiveness of the proposed AdvAD and AdvAD-X. Compared with state-of-the-art imperceptible attacks, AdvAD achieves an average of 99.9% (+17.3%) ASR with 1.34 (-0.97) $l_2$ distance, 49.74 (+4.76) PSNR and 0.9971 (+0.0043) SSIM against four prevalent DNNs with three different architectures on the ImageNet-compatible dataset. Code is available at https://github.com/XianguiKang/AdvAD.

## 1 Introduction

Deep Neural Networks (DNNs) are shown to be vulnerable to adversarial attacks [1, 2] (i.e., add maliciously crafted perturbations to the input data), posing serious security concerns to real-world applications [3]. The research of adversarial attacks also plays an important role in proactively exposing potential threats, as well as promoting model robustness and corresponding defense methods [4–11]. Many attacks [12–15] focus on maximizing the attack success rate and transferability under relatively lenient restrictions (i.e., $l_\infty$ or $l_2$ norm) of adversarial perturbation, but they could have poor stealthiness and imperceptibility since the crafted adversarial examples can be easily detected by the Human Visual System (HVS) [16]. Therefore, imperceptible adversarial attacks [17–23], aiming to maintain attacking efficacy while improving imperceptibility, have attracted considerable attention.

Current imperceptible adversarial attacks could be summarized into two categories: 1) perturbation-based attacks devised on perceptual characteristics, and 2) unrestricted attacks. The first one is motivated by the fact that adding adversarial perturbations to different components of an image has

---

*Corresponding author.

varying perceptual quality levels to the HVS. By studying components such as image color [19], texture complexity [21], frequency spectrum [23, 24], etc., these methods design corresponding perceptual-based loss functions and incorporate them to the optimization process to craft adversarial examples where the adversarial perturbation is constrained and hidden within specific image regions. Instead of injecting noise-like adversarial perturbations, unrestricted attacks heavily but reasonably modify attributes of images like semantic content to perform attacks. Apart from early work that adopts GANs [25], some recent methods combine the prevalent diffusion models [26–28] into the adversarial optimization process in an image edition-like way of repeatedly adding noise and denoising to eliminate the noise pattern within the final adversarial examples [29, 30] or optimize the embedding of latent diffusion models [31, 32]. However, due to the uncertainty of generative models and the unrestricted setting itself, some unrestricted adversarial examples inevitably exhibit obvious unnatural texture or semantic changes and lose the imperceptibility, especially for images with complex content. Although previous methods have equipped attacks with imperceptibility utilizing various designs mentioned above, it remains an essential challenge of achieving imperceptible adversarial attacks: *How to attack with inherently minimal perturbation strength from a modeling perspective?*

To address this fundamental challenge, we propose **Adversarial Attacks in Diffusion** (**AdvAD**), a brand new modeling framework distinct from common attack paradigms of gradient ascending [2] or optimization with adversarial losses [17]. The proposed AdvAD explores a novel *non-parametric* diffusion process for attacks, which fully inherits two key merits of diffusion models: **i)** the modeling philosophy of converting a difficult task into a series of simple sub-tasks, and **ii)** solid theoretical foundation. Specifically, AdvAD achieves high attack efficacy with intrinsically lower perturbation strength by innovatively modeling the attack process as a decomposed diffusion trajectory from an initialized noise to an adversarial example. At each step, a much subtler (for imperceptibility) yet more effective (for attack performance) *adversarial guidance* is calculated and injected with two cooperating, theoretically grounded non-parametric modules called Attacked Model Guidance (AMG) and Pixel-level Constraint (PC), which gradually leads the end of this trajectory from the original image distribution to a desired adversarially conditioned distribution based on the theory of diffusion models (e.g., deterministic diffusion[27], conditional sampling [33, 34], etc.).

Here, we would like to clarify that the proposed diffusion process for attacks is considered as non-parametric since it does not require additional networks as needed in regular diffusion models for noise estimation. AdvAD firstly initializes a fixed diffusion noise, which is then ingeniously manipulated at each step via the adversarial guidance crafted by the proposed AMG and PC modules using only the attacked model with theoretically derived equations. In this way, the proposed AdvAD is facilitated with the modeling approach of diffusion models rather than their denoising or generative capabilities, which avoids the negative impact like semantic content changes caused by the uncertainty of generative models and also promises relatively low computational complexity. Based on AdvAD, we further propose an enhanced version **AdvAD-X** ('X' for 'eXtreme') with two extra strategies to squeeze the extreme performance in an ideal scenario of the proposed new modeling framework with unique properties, which also possesses theoretical significance and provides new insights for revealing the robustness of DNNs. In summary, our main contributions are:

- Addressing the essential challenge of imperceptible adversarial attacks from a novel modeling perspective for the first time, we theoretically explore and derive the basic modeling of diffusion models to perform attacks with inherently lower perturbation strength through a non-parametric diffusion process that requires no additional networks.

- We propose two attack versions, AdvAD and AdvAD-X. For the basic AdvAD, the AMG and PC modules cooperate to craft much subtler yet effective adversarial guidance which is progressively injected via initialized diffusion noise at each step, and AdvAD-X further reduces the perturbation strength to an extreme level in an ideal scenario with theoretical significance.

- Extensive experiments are conducted to evaluate the effectiveness of our methods in terms of attack success rate, imperceptibility, and robustness. Experimental results demonstrate the superiority of the novel modeling approach for imperceptible adversarial attacks.

## 2   Preliminaries

**Adversarial Attacks.** Given an original image $x_{ori}$ with ground-truth label $y_{gt}$ and a classifier $f(\cdot)$ satisfying $f(x_{ori}) = y_{gt}$, normal untargeted attacks aim to craft the adversarial example $x_{adv}$ that

misleads the classifier, formulated as:

$$f(\boldsymbol{x}_{adv}) \neq y_{gt}, \quad s.t. \|\boldsymbol{x}_{adv} - \boldsymbol{x}_{ori}\|_p \leq \zeta, \tag{1}$$

where $\| \cdot \|_p$ represents $l_p$-norm that is usually implemented with $l_\infty$-norm to limit the distance between $\boldsymbol{x}_{adv}$ and $\boldsymbol{x}_{ori}$ within an upper bound of budget $\zeta$. In this paper, we focus on the more general setting of untargeted attacks. More information on related works is provided in **Appendix A**.

**Deterministic Diffusion Process.** In the deterministic situation of DDIM [27] with $\sigma_t = 0$, for an image $\boldsymbol{x}_0$ and pre-defined diffusion coefficients $\alpha_{0:T} \in (0, 1]^T$ for step $t \in [0 : T]$, $\boldsymbol{x}_t$ in the *Forward* process of adding noise to $\boldsymbol{x}_0$ is given by $\boldsymbol{x}_t = \sqrt{\alpha_t}\boldsymbol{x}_0 + \sqrt{1 - \alpha_t}\boldsymbol{\epsilon}$, where $\boldsymbol{\epsilon} \sim \mathcal{N}(\mathbf{0}, \boldsymbol{I})$ represents Gaussian noise. For the *Backward* denoising steps, unlike the DDPM [26] based on Markov chains that each state directly depends on the previous one, DDIM employs a non-Markovian approach. In the backward process, each step first involves calculating a "prediction" of final step $\boldsymbol{x}_t^0$ from current $\boldsymbol{x}_t$, then adding noise to it again to obtain $\boldsymbol{x}_{t-1}$, expressed as:

$$\boldsymbol{x}_{t-1} = \sqrt{\alpha_{t-1}}\left(\frac{\boldsymbol{x}_t - \sqrt{1 - \alpha_t}\boldsymbol{\epsilon}_\theta(\boldsymbol{x}_t)}{\sqrt{\alpha_t}}\right) + \sqrt{1 - \alpha_{t-1}}\boldsymbol{\epsilon}_\theta(\boldsymbol{x}_t), \tag{2}$$

where $\boldsymbol{\epsilon}_\theta(\boldsymbol{x}_t)$ is a estimated diffusion noise using a pretrained neural network $\theta$ for current step, and the term in the first parenthesis represents the predicted $\boldsymbol{x}_t^0$, derived by a simple variation of Eq. (2).

**Conditional sampling.** Song et al. [34] propose the conditional sampling technique for the *score-based* generative models with *score function* $\nabla_{\boldsymbol{x}_t}\log p(\boldsymbol{x}_t)$ [35], a kind of generative model has close relationship to diffusion models. Without loss of generality, for a condition $y$ (e.g., class label, mask, etc.) and corresponding conditional distribution $p(\boldsymbol{x}|y)$, a score-based model can sample from $p(\boldsymbol{x}|y)$ by modifying the score function at each step of $t$ to $\nabla_{\boldsymbol{x}_t}\log(p(\boldsymbol{x}_t)p(y|\boldsymbol{x}_t))$ if $p(y|\boldsymbol{x}_t)$ is known. Subsequently, with the connection between the score function and the noise $\boldsymbol{\epsilon}_t$ of diffusion models as $\nabla_{\boldsymbol{x}_t}\log p(\boldsymbol{x}_t) = -1/\sqrt{1 - \alpha_t}\boldsymbol{\epsilon}_t$ [34], this joint distribution could be expanded to the deterministic process of DDIM, achieved by updating the noise $\boldsymbol{\epsilon}_t$ to $\boldsymbol{\epsilon}_t'$ at each step as [33]:

$$\boldsymbol{\epsilon}_t' = \boldsymbol{\epsilon}_t - \sqrt{1 - \alpha_t}\nabla_{\boldsymbol{x}_t}\log p(y|\boldsymbol{x}_t). \tag{3}$$

## 3 Proposed Adversarial Attacks in Diffusion

### 3.1 Overview

From a novel modeling perspective, we propose **Adversarial Attacks in Diffusion (AdvAD)** to attack with inherently smaller perturbation strength through a non-parametric diffusion process for the first time. As shown in Figure 1, different from previous attack paradigms that employ gradient ascending or optimization with varying kinds of adversarial losses, AdvAD innovatively performs attack within a decomposed non-parametric diffusion trajectory starting from an initialized noise, in which very subtle yet effective adversarial guidance is crafted and injected to gradually push the end of this trajectory to a desired adversarially conditioned distribution from the original image.

Intuitively, given the original image $\boldsymbol{x}_{ori}$ with an initialized Gaussian noise $\boldsymbol{\epsilon}_0 \sim \mathcal{N}(\mathbf{0}, \boldsymbol{I})$, a fixed diffusion trajectory from $\bar{\boldsymbol{x}}_T$ to $\bar{\boldsymbol{x}}_0$ ($\bar{\boldsymbol{x}}_0 = \boldsymbol{x}_{ori}$) can be easily obtained using DDIM *Backward* for the deterministic diffusion process as:

$$\bar{\boldsymbol{x}}_T = \sqrt{\alpha_T}\boldsymbol{x}_{ori} + \sqrt{1 - \alpha_T}\boldsymbol{\epsilon}_0, \tag{4}$$

$$\bar{\boldsymbol{x}}_{t-1} = \sqrt{\alpha_{t-1}}\left(\frac{\bar{\boldsymbol{x}}_t - \sqrt{1 - \alpha_t}\boldsymbol{\epsilon}_0}{\sqrt{\alpha_t}}\right) + \sqrt{1 - \alpha_{t-1}}\boldsymbol{\epsilon}_0. \tag{5}$$

With this deterministic diffusion trajectory of the original image, performing adversarial attacks within it requires solving two main problems: **i)** directing the final result of this diffusion process to a desired adversarial example rather than the original image; **ii)** ensuring the modified trajectory (denoted as $\hat{\boldsymbol{x}}_t$, $\hat{\boldsymbol{\epsilon}}_t$ for step $t$) close to the original trajectory ($\bar{\boldsymbol{x}}_t$, $\boldsymbol{\epsilon}_0$ for step $t$) of the clean image to achieve the imperceptibility of attacks. To fulfill the dual purposes, we propose two theoretically grounded modules, called **Attacked Model Guidance (AMG)** and **Pixel-level Constraint (PC)** to work together. At each step, AMG utilizes only the attacked model $f(\cdot)$ to produce the adversarial guidance without requiring any additional networks, synergistically collaborated with PC to constrain and streamline the diffusion process injected with the guidances.

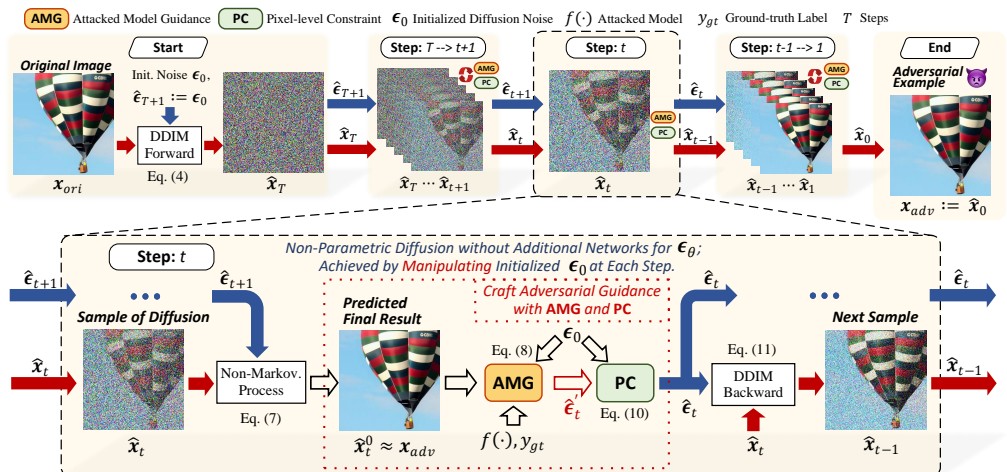

Figure 1: Overview of the proposed Adversarial Attacks in Diffusion (AdvAD) that models the attack as a non-parametric diffusing process. At each step, Attacked Model Guidance (AMG) module adopts the non-Markovian process for approximating $x_{adv}$ using $\hat{x}_t^0$ to craft adversarial guidance and injects it into the initialized diffusion noise, then Pixel-level Constraint (PC) module imposes restriction to produce the noise for the next step and serves to control the whole process precisely.

### 3.2 Attacked Model Guidance Module

By viewing the attack process as a distribution-to-distribution transformation through a non-parametric diffusion process, the proposed AMG module theoretically integrates the conditional sampling technique of diffusion models to craft the adversarial guidance only using the attacked model $f(\cdot)$. For untargeted attacks, the ultimate goal is modifying $x_{ori}$ with $f(x_{ori}) = y_{gt}$ to $x_{adv}$ so that $f(x_{adv}) \neq y_{gt}$, which can be regarded as directing the determined distribution $p(x_{ori})$ of the original diffusion trajectory to an distribution of $x_{adv}$ with the attacked model as $p(x_{adv}|f(x_{adv}) \neq y_{gt})$. Thus, we regard $f(x_{adv}) \neq y_{gt}$ as an adversarial condition, and employ the conditional sampling technique to the original trajectory by manipulating the diffusion noise to achieve this, expressed as:

$$\hat{\epsilon}_t' = \epsilon_0 - \sqrt{1 - \alpha_t}\nabla_{\hat{x}_t}\log p(f(x_{adv}) \neq y_{gt}|\hat{x}_t)$$
$$= \epsilon_0 - \sqrt{1 - \alpha_t}\nabla_{\hat{x}_t}\log(1 - p(f(x_{adv}) = y_{gt}|\hat{x}_t)). \tag{6}$$

However, Eq. (6) is unsolvable since $x_{adv}$ is unknown during the diffusing process. To address this, inspired by the properties of deterministic non-Markovian DDIM that a final diffusion result is firstly predicted at each step, we calculate $\hat{x}_t^0$ via the equation of DDIM non-Markovian process with $\hat{\epsilon}_{t+1}$ from the previous step, and use it to approximate $x_{adv}$, expressed as:

$$x_{adv} \approx \hat{x}_t^0 = \frac{\hat{x}_t - \sqrt{1 - \alpha_t}\,\hat{\epsilon}_{t+1}}{\sqrt{\alpha_t}}. \tag{7}$$

The accurate error upper bound and convergence of this approximation are given in Proposition 2 in conjunction with the proposed PC module, and the validity of this approximation can also be explained intuitively from the premise of our method. That is, we have $\bar{x}_t^0 = x_{ori}$ for all step $t$ in the original diffusion trajectory, and the modified trajectory should be very close to the original one, so that the relationship between $\hat{x}_t^0$ and $x_{adv}$ should satisfy $\hat{x}_t^0 \approx x_{adv}$.

With Eq. (7), the term of $p(f(x_{adv}) = y_{gt}|\hat{x}_t)$ in Eq. (6) can be written as $p(f(\hat{x}_t^0) = y_{gt}|\hat{x}_t) = p(f(\hat{x}_t^0) = y_{gt})$ since $\hat{x}_t^0$ is calculated from $\hat{x}_t$, which is exactly the output logits of $f(\hat{x}_t^0)$ with $Softmax(\cdot)$ function for the class $y_{gt}$. Denoting this term as the classification probability of the attacked model as $p_f(y_{gt}|\hat{x}_t^0)$, we can obtain the solvable equation of AMG module that injects adversarial guidance to the initialized diffusion noise using only $f(\cdot)$ without any additional network:

$$\hat{\epsilon}_t' = \text{AMG}(\epsilon_0, \hat{x}_t^0, f(\cdot), y_{gt}) = \epsilon_0 - \sqrt{1 - \alpha_t}\nabla_{\hat{x}_t}\log(1 - p_f(y_{gt}|\hat{x}_t^0)). \tag{8}$$

At this point, in addition to the benefits from modeling, this calculation process of AMG also plays a role in endowing AdvAD with imperceptibility. As the attack progresses, the probability $p_f$, the term

---
**Algorithm 1** AdvAD
---
**Input**: Attacked model $f(\cdot)$, image $\boldsymbol{x}_{ori}$ with label $y_{gt}$, budget $\xi$, step $T$;
**Output**: Adversarial example $\boldsymbol{x}_{adv}$

 1: Initialize pre-defined diffusion coefficients $\alpha_{0:T} \in (0, 1]^{T+1}$;
 2: Initialize $\boldsymbol{\epsilon}_0 \sim \mathcal{N}(\boldsymbol{0}, \boldsymbol{I})$;          ▷ Initialize and fix diffusion noise $\boldsymbol{\epsilon}_0$.
 3: Transform the range of $\boldsymbol{x}_{ori}$ to [-1, 1];     ▷ Align with data range of diffusion process.
 4: Calculate $\bar{\boldsymbol{x}}_T$ via Eq. (4);       ▷ Forward process of adding noise $\boldsymbol{\epsilon}_0$ to $\boldsymbol{x}_{ori}$.
 5: Set $\hat{\boldsymbol{x}}_T := \bar{\boldsymbol{x}}_T$, $\hat{\boldsymbol{\epsilon}}_{T+1} := \boldsymbol{\epsilon}_0$;       ▷ Non-parametric diffusion process.
 6: **for** $t = T$ to 1 **do**
 7:   Calculate $\hat{\boldsymbol{x}}_t^0$ via Eq. (7);       ▷ Approximation of $\hat{\boldsymbol{x}}_t^0 \approx \boldsymbol{x}_{adv}$.
 8:   Transform the range of $\hat{\boldsymbol{x}}_t^0$ to [0, 255];     ▷ Align with data range of image.
 9:   Calculate $\hat{\boldsymbol{\epsilon}}_t'$ with AMG via Eq. (8);      ▷ Inject adversarial guidance.
10:   Calculate $\hat{\boldsymbol{\epsilon}}_t$ with PC via Eq. (10);     ▷ Constraint modified diffusion noise.
11:   Calculate $\hat{\boldsymbol{x}}_{t-1}$ via Eq. (11);     ▷ One step backward from $t$ to $t-1$.
12: Transform the range of $\hat{\boldsymbol{x}}_0$ to [0, 255];       ▷ Endpoint of the process.
13: **return** $\boldsymbol{x}_{adv} = \text{int8(round}(\hat{\boldsymbol{x}}_0))$;      ▷ Return actual 8-bit image $\boldsymbol{x}_{adv}$.
---

of $\log(1 - p_f)$ as well as coefficient $\sqrt{1 - \alpha_t}$ gradually approach 0, which means the strength of injected adversarial guidance gradually converge to 0 in AdvAD, while common classification losses (e.g, Cross-Entropy, Log Loss, etc.) used in other attack paradigms may increase on the contrary. Further analysis and experiments on this property are provided in Proposition 1 and Sec. 4.5.

### 3.3 Pixel-level Constraint Module

Collaborating with AMG, the PC module is introduced to impose precise control and streamline the modified diffusion trajectory for attacks. A straightforward choice is to design PC for $\hat{\boldsymbol{x}}_t$ that constrains each $\hat{\boldsymbol{x}}_t$ using $\bar{\boldsymbol{x}}_t$, thus ensuring $\hat{\boldsymbol{x}}_t^0$ close to $\bar{\boldsymbol{x}}_t^0$ and the final $\boldsymbol{x}_{adv}$ close to $\boldsymbol{x}_{ori}$. However, such a "hard" constraint directly applied to $\hat{\boldsymbol{x}}_t$ will impair the effectiveness of AMG and disrupt coherence of the transforming trajectory. Therefore, we formulate a more suitable PC for $\hat{\boldsymbol{\epsilon}}_t$ as in Theorem 1.

**Theorem 1** *Given diffusion coefficients $\alpha_{T:0} \in (0, 1]^T$, the $\boldsymbol{x}_{ori}$, $\bar{\boldsymbol{x}}_t$, $\boldsymbol{\epsilon}_0$ from the original trajectory, $\hat{\boldsymbol{x}}_t$, $\hat{\boldsymbol{\epsilon}}_t$ from the modified trajectory, and a variable $\xi$, if $\hat{\boldsymbol{\epsilon}}_t$ and $\boldsymbol{\epsilon}_0$ satisfies*

$$\|\hat{\boldsymbol{\epsilon}}_t - \boldsymbol{\epsilon}_0\|_\infty \leq \frac{\sqrt{\alpha_T}}{\sqrt{1 - \alpha_T}}\xi, \tag{9}$$

*for all $t \in [T:1]$, then it follows that $\|\hat{\boldsymbol{x}}_t - \bar{\boldsymbol{x}}_t\|_\infty \leq (\sqrt{\alpha_t} - \sqrt{1 - \alpha_t}\frac{\sqrt{\alpha_T}}{\sqrt{1-\alpha_T}})\xi$, $\|\hat{\boldsymbol{x}}_t^0 - \boldsymbol{x}_{ori}\|_\infty \leq \xi$, and $\|\hat{\boldsymbol{x}}_0 - \boldsymbol{x}_{ori}\|_\infty \leq \xi$ hold true.*

According to Theorem 1, the PC for $\hat{\boldsymbol{\epsilon}}_t$ is implemented as:

$$\hat{\boldsymbol{\epsilon}}_t = \text{PC}(\hat{\boldsymbol{\epsilon}}_t') = \mathcal{P}_{l_\infty(\boldsymbol{\epsilon}_0, \frac{\sqrt{\alpha_T}}{\sqrt{1-\alpha_T}}\xi)}(\hat{\boldsymbol{\epsilon}}_t'). \tag{10}$$

where $\mathcal{P}_{l_\infty(\boldsymbol{\epsilon}, \xi)}(\cdot)$ is a projection operation that constrains the output $\hat{\boldsymbol{\epsilon}}_t'$ of AMG$(\cdot)$ to $\hat{\boldsymbol{\epsilon}}_t$ based on a $l_\infty$-norm ball of $\boldsymbol{\epsilon}_0$ to satisfy Eq. (9). After PC, the diffusion noise $\hat{\boldsymbol{\epsilon}}_t$ for next step is obtained, and the $\hat{\boldsymbol{x}}_{t-1}$ can be calculated using the deterministic DDIM *backward* equation as:

$$\hat{\boldsymbol{x}}_{t-1} = \sqrt{\alpha_{t-1}}\left(\frac{\hat{\boldsymbol{x}}_t - \sqrt{1 - \alpha_t}\hat{\boldsymbol{\epsilon}}_t}{\sqrt{\alpha_t}}\right) + \sqrt{1 - \alpha_{t-1}}\hat{\boldsymbol{\epsilon}}_t. \tag{11}$$

The elaborate PC for $\hat{\boldsymbol{\epsilon}}_t$ directly cooperates with AMG to constrain the diffusion noise, which streamlines the whole diffusion process and can serve to simultaneously control the terms of $\hat{\boldsymbol{x}}_t$, $\hat{\boldsymbol{x}}_t^0$, and $\hat{\boldsymbol{x}}_0$, satisfying the premise that two trajectories are close and ensuring the effectiveness of AdvAD. The complete pseudo code of AdvAD is provided in Algorithm 1.

Subsequently, based on Theorem 1, we further give two propositions about AdvAD as:

**Proposition 1** *Under the conditions of Theorem 1, by denoting constrained $\hat{\boldsymbol{\epsilon}}_t = \boldsymbol{\epsilon}_0 - \boldsymbol{\delta}_t$, we have*

$$\boldsymbol{x}_{adv} = \boldsymbol{x}_{ori} + \sum_{t=1}^{T} \lambda_t \boldsymbol{\delta}_t, \tag{12}$$

where $\lambda_t = \frac{\sqrt{1-\alpha_t}}{\sqrt{\alpha_t}} - \frac{\sqrt{1-\alpha_{t-1}}}{\sqrt{\alpha_{t-1}}}$, and $\|\boldsymbol{\delta}_t\|_\infty \leq \frac{\sqrt{\alpha_T}}{\sqrt{1-\alpha_T}}\xi$.

**Proposition 2** *Under the conditions of Theorem 1, the upper bound on the error of the approximation in Eq.* (7) *can be expressed as*

$$\left\|\boldsymbol{x}_{adv} - \hat{\boldsymbol{x}}_t^0\right\|_\infty \leq 2 \cdot \frac{\sqrt{1-\alpha_t}}{\sqrt{\alpha_t}}\frac{\sqrt{\alpha_T}}{\sqrt{1-\alpha_T}}\xi. \tag{13}$$

Proposition 1 explicitly states the much subtler and decreasing strength of the adversarial guidance injected at each step of AdvAD's non-parametric diffusion process, and also allows for a quantitative analysis (as in Sec. 5.5). Proposition 2 indicates the validity and convergence of the approximation of $\boldsymbol{x}_{adv} \approx \hat{\boldsymbol{x}}_t^0$ in AMG (Eq. (7)). It is evident that as $t$ goes from $T$ to 1, $\alpha_t$ increases from 0 to 1, the upper bound on the approximation error rapidly converge from $2\xi$ to 0. The detailed derivations of the mentioned PC for $\hat{\boldsymbol{x}}_t$, proofs of Theorem 1 and Proposition 1, 2 are provided in **Appendix B**.

### 3.4 AdvAD to AdvAD-X: Extreme Version

Building upon AdvAD, we further propose a scheme called AdvAD-X ('X' for 'eXtreme') with two extra strategies called Dynamic Guidance Injection (DGI) and CAM Assistance (CA), aiming to squeeze the extreme performance of our novel modeling framework in an ideal scenario that is usually overlooked but has theoretical significance.

**DGI and CA Strategies.** As aforementioned, the attack capability of AdvAD comes from the very subtle yet effective adversarial guidance crafted by AMG and PC, and the intensity of guidance will decrease to 0 as the process progresses. Thus, the DGI is naturally emerged as a dynamic skipping strategy to skip the unnecessary calculation and injection of adversarial guidance, especially for those steps in the later process. With DGI, AdvAD-X dynamically avoids the execution of AMG and PC and adopts original $\boldsymbol{\epsilon}_0$ as the diffusion noise for the steps where the $\hat{\boldsymbol{x}}_t^0 \approx \boldsymbol{x}_{adv}$ is already able to mislead the attacked model, reducing the accumulated guidance strength as well as the computational complexity. On the other hand, inspired by the Class Activation Mapping (CAM) [36] identifies critical regions of an image about a decision made of a classifier, our CA strategy calculates a mask (if available) $\boldsymbol{m}$ ranging from 0 to 1 of $\boldsymbol{x}_{ori}$ with $f(\cdot)$ and $y_{gt}$ using GradCAM [37] to further suppress the strength of adversarial guidance within the non-critical image regions in those steps that are not skipped. The equation of AMG with CA strategy can be modified as:

$$\hat{\boldsymbol{\epsilon}}_t' = \boldsymbol{\epsilon}_0 - \boldsymbol{m} \cdot \sqrt{1-\alpha_t}\nabla_{\hat{\boldsymbol{x}}_t}\log(1 - p_f(y_{gt}|\hat{\boldsymbol{x}}_t^0)). \tag{14}$$

**Ideal Scenario.** Equipped with DGI, AdvAD-X omits a large number of adversarial guidance that is injected by default in AdvAD, while the absolute strength of guidance in each of the remaining steps are also suppressed by CA, successfully reducing the final adversarial perturbation to an extreme level. This extreme case leads to a problem that in the default setting of attacking with 8-Bit RGB images, the adversarial perturbation of pixels where the intensity is less than $0.5$ will be erased due to the quantization. However, in practice, the input of DNNs is normalized as floating-point data type to avoid gradient problems during training [38, 39], and white-box attack allows access to the entire of DNNs. Therefore, for AdvAD-X, we specifically consider an ideal scenario that directly input the raw final adversarial example in floating-point data to DNNs without quantization to evaluate the extreme performance of AdvAD-X. The pseudo code of AdvAD-X is provided in **Appendix C**.

## 4 Experiments

### 4.1 Experimental Setup

**Dataset.** In line with prior studies [15, 19, 32, 40], our experiments are conducted on the ImageNet-compatible Dataset [1], containing 1,000 images of ImageNet [41] classes with size of $299 \times 299$, and the images are resized to standard input size of $224 \times 224$ in all experiments. **Models.** We select the widely used CNNs of ResNet-50 [42] and enhanced ConvNeXt-Base [43], Swin Transformer-Base

---

[1] https://github.com/cleverhans-lab/cleverhans/tree/master/cleverhans_v3.1.0/examples/nips17_adversarial_competition/dataset

Table 1: Results of untargeted white-box attack success rate (ASR) and other evaluation metrics for imperceptibility when employing different attacks and attacked models. The reported running times are obtained using a RTX 3090 GPU on a same machine. † and blue mean the results of AdvAD-X are obtained with floating-point data type in the ideal scenario as described in Sec 3.4.

| Model | Attack Method | Time (s) ↓ | ASR (%) ↑ | $l_\infty$ ↓ | $l_2$ ↓ | PSNR ↑ | SSIM ↑ | FID ↓ | LPIPS ↓ | MUSIQ ↑ |
|---|---|---|---|---|---|---|---|---|---|---|
| ResNet-50 [42] | PGD [12] | 25 | 98.6 | 0.031 | 8.17 | 33.53 | 0.8830 | 35.25 | 0.0517 | 52.24 |
| | NCF [40] | 2739 | 89.9 | 0.783 | 75.16 | 14.79 | 0.6374 | 58.99 | 0.3052 | 49.12 |
| | ACA [32] | 82239 | 89.8 | 0.839 | 52.42 | 18.00 | 0.5659 | 69.57 | 0.3381 | 55.47 |
| | DiffAttack [31] | 34954 | 96.6 | 0.743 | 30.51 | 22.63 | 0.6750 | 55.29 | 0.1130 | 55.67 |
| | DiffPGD [30] | 6057 | 92.1 | 0.246 | 11.43 | 30.95 | 0.8902 | 22.18 | 0.0315 | 55.05 |
| | AdvDrop [21] | 193 | 96.8 | 0.062 | 3.17 | 41.91 | 0.9872 | 5.57 | 0.0061 | 54.96 |
| | PerC-AL [19] | 4085 | 98.8 | 0.131 | 2.05 | 46.35 | 0.9894 | 8.62 | 0.0029 | 55.84 |
| | SSAH [24] | 428 | 99.7 | 0.033 | 2.65 | 43.73 | 0.9911 | 4.48 | 0.0021 | 55.49 |
| | AdvAD (ours) | 2201 | 99.7 | 0.010 | 1.06 | 51.84 | 0.9980 | 2.42 | 0.0005 | 56.35 |
| | AdvAD-X† (ours) | 806 | 100.0 | 0.002 | 0.34 | 63.62 | 0.9997 | 0.23 | 0.0001 | 56.59 |
| ConvNeXt-Base [43] | PGD [12] | 127 | 99.9 | 0.031 | 7.98 | 33.74 | 0.8845 | 32.03 | 0.0386 | 51.85 |
| | NCF [40] | 5222 | 59.4 | 0.750 | 72.89 | 15.10 | 0.6616 | 50.52 | 0.2846 | 49.70 |
| | ACA [32] | 83149 | 82.2 | 0.835 | 52.16 | 18.05 | 0.5676 | 68.45 | 0.3421 | 55.11 |
| | DiffAttack [31] | 35417 | 97.8 | 0.754 | 31.70 | 22.28 | 0.6610 | 72.22 | 0.1277 | 54.80 |
| | DiffPGD [30] | 6325 | 76.9 | 0.245 | 11.45 | 30.94 | 0.8908 | 21.05 | 0.0306 | 54.75 |
| | AdvDrop [21] | 838 | 96.9 | 0.057 | 3.26 | 41.69 | 0.9864 | 6.42 | 0.0055 | 54.80 |
| | PerC-AL [19] | 18271 | 10.3 | - | - | - | - | - | - | - |
| | SSAH [24] | 3423 | 84.6 | 0.026 | 2.24 | 45.19 | 0.9928 | 3.04 | 0.0011 | 55.78 |
| | AdvAD (ours) | 15240 | 100.0 | 0.016 | 1.49 | 48.61 | 0.9964 | 5.07 | 0.0009 | 55.97 |
| | AdvAD-X† (ours) | 5245 | 99.8 | 0.004 | 0.64 | 58.01 | 0.9993 | 0.62 | 0.0001 | 56.43 |
| Swin Trans.-Base [44] | PGD [12] | 93 | 98.5 | 0.031 | 7.85 | 33.88 | 0.8861 | 21.34 | 0.0378 | 51.91 |
| | NCF [40] | 4690 | 63.7 | 0.733 | 69.92 | 15.48 | 0.6822 | 47.17 | 0.2709 | 49.77 |
| | ACA [32] | 83706 | 79.6 | 0.831 | 50.70 | 18.31 | 0.5757 | 64.83 | 0.3341 | 55.65 |
| | DiffAttack [31] | 36736 | 89.7 | 0.741 | 30.45 | 22.67 | 0.6727 | 53.32 | 0.1143 | 55.72 |
| | DiffPGD [30] | 6499 | 69.1 | 0.244 | 11.26 | 31.10 | 0.8945 | 16.19 | 0.0276 | 55.25 |
| | AdvDrop [21] | 673 | 97.2 | 0.063 | 3.37 | 41.43 | 0.9853 | 5.22 | 0.0065 | 54.73 |
| | PerC-AL [19] | 15258 | 95.6 | 0.144 | 2.15 | 45.93 | 0.9882 | 3.53 | 0.0015 | 55.66 |
| | SSAH [24] | 1737 | 96.3 | 0.035 | 2.41 | 44.60 | 0.9927 | 2.57 | 0.0010 | 55.53 |
| | AdvAD (ours) | 9729 | 100.0 | 0.013 | 1.19 | 50.57 | 0.9978 | 1.70 | 0.0004 | 56.17 |
| | AdvAD-X† (ours) | 5243 | 99.7 | 0.005 | 0.52 | 60.29 | 0.9995 | 0.25 | 0.0001 | 56.47 |
| VisionMamba-Small [46] | PGD [12] | 63 | 95.7 | 0.031 | 7.99 | 33.73 | 0.8884 | 26.09 | 0.0503 | 52.37 |
| | NCF [40] | 3919 | 71.7 | 0.738 | 68.71 | 15.68 | 0.6876 | 46.07 | 0.2629 | 50.05 |
| | ACA [32] | 96851 | 84.2 | 0.831 | 50.88 | 18.28 | 0.5753 | 65.77 | 0.3329 | 55.28 |
| | DiffAttack [31] | 43043 | 90.9 | 0.749 | 30.94 | 22.52 | 0.6693 | 52.16 | 0.1179 | 55.66 |
| | DiffPGD [30] | 7638 | 83.4 | 0.248 | 11.75 | 30.68 | 0.8845 | 21.02 | 0.0378 | 54.19 |
| | AdvDrop [21] | 1311 | 97.0 | 0.076 | 4.42 | 39.30 | 0.9761 | 8.02 | 0.0086 | 54.34 |
| | PerC-AL [19] | 10400 | 6.5 | - | - | - | - | - | - | - |
| | SSAH [24] | 1204 | 49.8 | 0.028 | 1.95 | 46.41 | 0.9946 | 2.08 | 0.0018 | 55.96 |
| | AdvAD (ours) | 6154 | 99.7 | 0.016 | 1.62 | 47.94 | 0.9960 | 3.67 | 0.0017 | 56.17 |
| | AdvAD-X† (ours) | 4021 | 99.4 | 0.005 | 0.69 | 58.90 | 0.9989 | 0.51 | 0.0004 | 56.50 |

[44] with Transformer [45] architecture, and VisionMamba-Small [46] with the recently emerged advanced Mamba [47] architecture. **Attacks.** We choose classic PGD [12] and seven attacks that claim having imperceptibility as comparison methods, including normal imperceptible attacks of AdvDrop [21], PerC-AL [19], SSAH [24], and unrestricted attacks of NCF [40], ACA [32], DiffAttack [31], Diff-PGD [30], and the generative capability of diffusion models are utilized the last three attacks. For our proposed AdvAD and AdvAD-X, we set $\xi = 8/255$ and $T = 1000$ for all experiments unless specifically mentioned. All the other comparison methods are evaluated using their official open-source code with the default hyper-parameters. The results of AdvAD-X are obtained in the ideal scenario with float-pointing raw data as described in Sec. 3.4. **Evaluation Metrics.** Attack success rate (ASR) is used to evaluate the attack efficacy, and seven metrics are adopted to comprehensively assess the imperceptibility, including $l_2$ and $l_\infty$ distances for absolute perturbation strength; Peak Signal-to-Noise Ratio (PSNR), Structure Similarity (SSIM) [48], and three network-based metrics, i.e., Learned Perceptual Image Patch Similarity (LPIPS) [49], Fréchet Inception Distance (FID) [50], and a non-reference metric MUSIQ [51] for image quality.

## 4.2 Comparison with State-of-the-art Methods

**White-Box Attacks.** Table 1 reports the untargeted attack performance and imperceptibility of ten methods against four attacked models. It is evident that the proposed AdvAD with novel modeling framework consistently demonstrates superior performance in terms of both ASR and imperceptibility. For the normal imperceptible adversarial attacks, the absolute adversarial perturbation strength of AdvAD in $l_\infty$ and $l_2$ distance are only 0.014 and 1.34 in average, which is about half of the state-of-the-art restricted imperceptible attack SSAH, and AdvAD maintains almost 99.9% ASR, supporting our key idea that it inherently reduces the strength of perturbation required for attacks from a modeling perspective. When attacking more advanced models from ResNet to VisionMamba, AdvAD always demonstrates the best ASR and imperceptibility, yet other methods tend to have some performance degradation (e.g., PerC-AL and SSAH for ConvNeXt and VisionMamba). For unrestricted attacks,

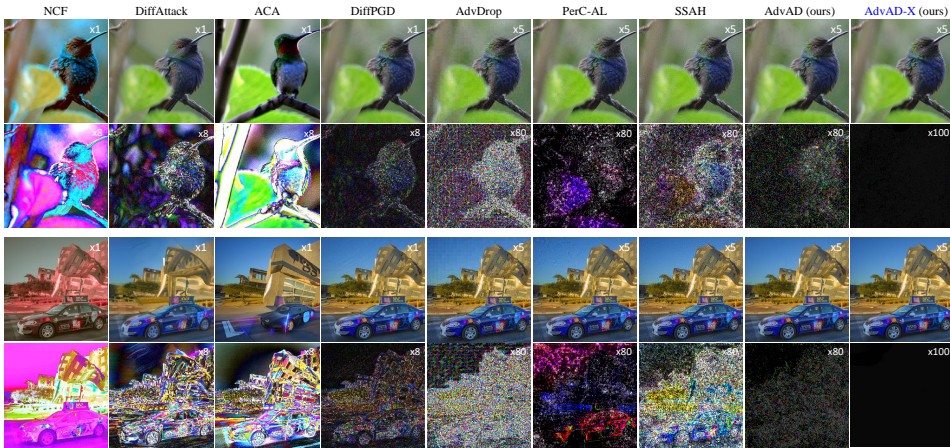

Figure 2: Visualizations of adversarial examples and corresponding perturbations crafted by nine imperceptible attacks. Perturbations are amplified as marked in top-right for the convenience of observation. Please zoom in to observe the details of the images with original resolution of $224 \times 224$.

Table 2: Results of ASR against defenses for robustness evaluation, including three post-processing purification methods and four adversarial training white-box robust models.

| Attack Method | Post Purifications (Normal Res-50) | | | | Attack Adversarial Training Model | | | | | All Avg. |
|---|---|---|---|---|---|---|---|---|---|---|
| | NRP [4] | DS [52] | Diffusion [5] | Avg. | Inc-V3 [8] | Res-50 [9] | Swin-B [53] | ConvNeXt-B [53] | Avg. | |
| AdvDrop [21] | 50.2 | **30.1** | **37.1** | 39.1 | 93.7 | 72.4 | 31.2 | 37.3 | 58.7 | 50.3 |
| PerC-AL [19] | 30.3 | 28.8 | 25.4 | 28.2 | **99.9** | 46.1 | 8.2 | 7.0 | 40.3 | 35.1 |
| SSAH [24] | 25.6 | 28.0 | 11.0 | 21.5 | 91.2 | **84.6** | 16.8 | 47.4 | 60.0 | 43.5 |
| AdvAD (ours) | **51.5** | 29.5 | 31.2 | 37.4 | 98.9 | 79.3 | 60.2 | 62.7 | 75.3 | **59.0** |
| AdvAD-X[†] (ours) | 13.4 | 27.6 | 10.2 | 17.1 | 57.2 | 45.2 | 18.0 | 16.2 | 34.2 | 26.8 |

it is expected for them to perform poorly in the quantitative metrics, but if the results are poor for all image quality metrics, it usually indicates that the images are damaged. Meanwhile, since the optimizer may not find the global optimal solution, the optimization-based methods tent to show sub-optimal ASR. For AdvAD-X, surprisingly, the perturbation strength is reduced to an extremely low level with still high attack efficacy in the ideal scenario with floating-point raw data.

**Visualization.** The visualizations of adversarial examples agint ResNet-50 in Figure 2 clearly show the characteristics of different imperceptible attacks against ResNet-50. For the first image with a relatively simple and clear object, the unrestricted attacks of NCF, DiffAttack and ACA perform attacks by modifying the semantics fairly, while DiffPGD uses denoising to avoid significant semantic modifications, but often has lower ASR as in Table 1. However, for the image with complex content, the unrestricted attacks result in obvious unnatural color, texture, artifacts and semantic changes. For the normal attacks with perceptual-based restrictions, by amplifying the noises, it can be seen that AdvDrop has a obvious gridding effect due to the blocking operation in DCT operation, and the perturbation strength in PerC-AL and SSAH is also related to the edge or texture components of the image. In contrast, our AdvAD continuously maintains uniform and lower perturbation which is very difficult to be seen even in the adversarial examples with ×5 noise. For AdvAD-X, the perturbations are very slight modifications to the decimal places of the floating-point raw data for each pixel, thus it is still difficult to be seen even after ×100 magnification. More quantitative comparisons and visualizations are provided in **Appendix** D.1, D.2.

## 4.3 Robustness

The robustness of attacks is also evaluated against defense methods, including purification methods of NRP [4], DS [52], diffusion-based purification [5] and adversarial training robust models of Inc-V3 [8], Res-50 [9], Swin-B [53], ConvNeXt-B [53]. Two classic image transformation defenses of JPEG compression [54], Bit-depth reduction [55], and another type of defense, random smoothing [11], are also included. Considering the robustness and transferability of attacks are comparable only under close perturbation budget, the unrestricted attacks are not included in this and the next section.

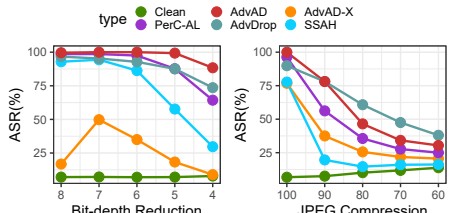

Figure 3: Rubostness on JPEG compression and Bit-depth reduction with different factors.

Table 3: Results of imperceptible attacks against random smoothing defense. Adversarial examples are crafted using only the base model, and then 100 rounds of random smoothing are applied to obtain the final ASR. $\sigma$ is the variance of smoothing noise.

|  | $\sigma = 0.25$ | | $\sigma = 0.50$ | | $\sigma = 1.00$ | |
| --- | --- | --- | --- | --- | --- | --- |
|  | ASR↑ | $l_2$↓ | ASR↑ | $l_2$↓ | ASR↑ | $l_2$↓ |
| clean | 17.3 | - | 30.3 | - | 46.8 | - |
| AdvDrop | 25.2 | 5.97 | 33.5 | 6.21 | 48.7 | 5.61 |
| SSAH | 21.8 | 13.84 | 32.4 | 14.82 | 46.9 | 13.68 |
| AdvAD (ours) | **28.2** | **2.41** | **36.8** | **2.51** | **50.4** | **2.08** |

Table 4: Transferability and effect of $T$ of the proposed AdvAD. $*$ means white-box ASR.

| Model | Attack Method | Res-50 | Mob-V2 | Inc-V3 | VGG-19 | $l_2$ ↓ | PSNR ↑ | SSIM ↑ | FID ↓ | LPIPS ↓ |
| --- | --- | --- | --- | --- | --- | --- | --- | --- | --- | --- |
| Res-50 [42] | SSAH [24] | 99.7* | 15.5 | 20.4 | 12.7 | 2.65 | 43.73 | 0.9911 | 4.48 | 0.0021 |
|  | AdvAD ($T$=1000) | 99.7* | **18.3** | **22.6** | **15.1** | **1.06** | **51.84** | **0.9980** | **2.42** | **0.0005** |
|  | AdvDrop [21] | 96.8* | 17.3 | 23.1 | 15.8 | 3.17 | 41.91 | 0.9872 | **5.57** | 0.0061 |
|  | PerC-AL [19] | 98.8* | 22.4 | 23.8 | 17.4 | 2.05 | **46.35** | 0.9894 | 8.62 | 0.0029 |
|  | AdvAD ($T$=100) | **100.0*** | **23.5** | **24.9** | **19.9** | **1.97** | 46.04 | **0.9912** | 7.15 | **0.0026** |
|  | PGD [12] | 98.6* | 41.4 | 36.7 | 36.0 | 8.17 | 33.53 | 0.8830 | 35.25 | **0.0517** |
|  | AdvAD ($T$=10) | **100.0*** | **44.3** | **37.6** | **42.9** | **7.21** | **34.63** | **0.9015** | **30.84** | 0.0547 |
| Mob-V2 [56] | SSAH [24] | 7.7 | 97.8* | 19.8 | 11.6 | 2.18 | 45.24 | 0.9930 | 2.95 | 0.0016 |
|  | AdvAD ($T$=1000) | **9.7** | **99.7*** | **21.3** | **14.8** | **0.94** | **53.08** | **0.9982** | **1.46** | **0.0004** |
|  | AdvDrop [21] | 9.7 | 97.7* | 22.7 | 15.0 | 3.16 | 41.94 | 0.9873 | 4.88 | 0.0064 |
|  | PerC-AL [19] | **12.7** | 99.8* | 23.3 | 17.8 | 2.16 | 45.67 | 0.9879 | 8.77 | 0.0032 |
|  | AdvAD ($T$=100) | 12.2 | **100.0*** | **23.4** | **17.9** | **1.83** | **46.68** | **0.9919** | **4.73** | **0.0020** |
|  | PGD [12] | 29.9 | 99.9* | **35.3** | 37.9 | 8.29 | 33.41 | 0.8803 | 34.57 | 0.0500 |
|  | AdvAD ($T$=10) | **30.6** | **100.0*** | **35.3** | **38.5** | **7.23** | **34.60** | **0.9006** | **27.25** | **0.0480** |

As shown in Table 2, the proposed AdvAD demonstrates the best robustness in overall average compared with other imperceptible attacks of AdvDrop, PerC-AL and SSAH. Specifically, when attacking robust models, AdvAD achieved an much higher average ASR of 75.3%. For post-processing purifications aim at eliminating adversarial perturbations, despite the inherently lower perturbation strength, AdvAD still maintains the best or second-best ASR against different purifications, which is comparable to AdvDrop with much higher perturbation strength. Similarly, for the results of classic image transformation defenses in Figure 3, AdvAD also exhibits advantages in most of the factors. In addition, since random smoothing is not a truly end-to-end method but a method that uses the base model to make multiple predictions on noise-augmented images, we adopt a semi-white-box setup to fully test the attack performance as described in the caption. Table 3 shows the experimental results, and the PerC-AL is not included because it fails to attack in this setting. It can be seen that for all $\sigma$, our AdvAD continuously achieves the best ASR with smaller perturbation strength.

We suppose the robustness of AdvAD mainly benefits from two aspects. Firstly, AdvAD performs attacks during a unique non-parametric diffusion process with adversarial guidance, which may be easier to break through existing adversarial training models using common attack paradigms. On the other hand, the inherently lower perturbation crafted by AdvAD is spread across the images more uniformly rather than gathering in some areas as can be seen in the visualization, making it more difficult to be eliminated. For AdvAD-X, it is anticipated to exhibit weak robustness since the extremely low perturbation in the ideal scenario is easy to defense.

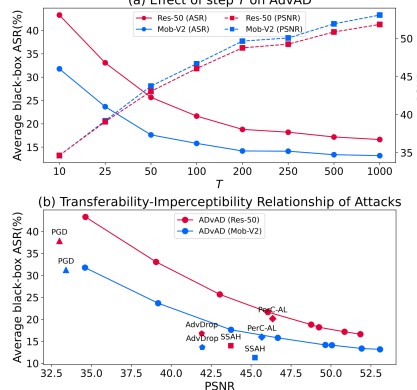

Figure 4: More results of (a) effect of step $T$ on AdvAD and (b) transferability-imperceptibility relationship of attacks.

### 4.4 Transferability and Effect of Step $T$ on AdvAD

Table 4 reports the ASRs of black-box attacks and the corresponding results of imperceptibility. We also test AdvAD with different step of $T$ for comprehensive evaluation. Consistent with the diffusion models, a larger $T$ denotes a finer decomposition granularity of the entire process, corresponding to the strength of adversarial guidance at each step. Thus, AdvAD with a larger $T$ exhibits better imperceptibility, while a smaller $T$ implies stronger black-box transferability. Notbly, though there is a clear negative correlation between imperceptibility and transferability, our

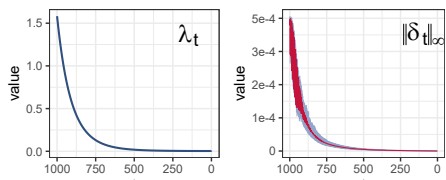

Figure 5: Values of $\lambda_t$ (left) and $\|\boldsymbol{\delta}_t\|_\infty$ (right) of Eq. (12) throughout the diffusion process.

Table 5: Results of AdvAD and AdvAD-X with smaller $\xi$ against Res-50. Step $T$ is fixed as 1000.

| $\xi$ | Attack | ASR↑ | $l_2$↓ | PSNR↑ | SSIM↑ | FID↓ |
|---|---|---|---|---|---|---|
| 4/255 | AdvAD | 98.6 | 0.93 | 53.27 | 0.9986 | 1.78 |
| | AdvAD-X[†] | 100.0 | 0.29 | 65.07 | 0.9998 | 0.18 |
| 2/255 | AdvAD | 96.1 | 0.82 | 54.85 | 0.9989 | 1.33 |
| | AdvAD-X[†] | 99.4 | 0.27 | 65.95 | 0.9998 | 0.15 |
| 1/255 | AdvAD | 87.4 | 0.66 | 57.87 | 0.9993 | 0.77 |
| | AdvAD-X[†] | 94.8 | 0.26 | 66.42 | 0.9998 | 0.14 |

AdvAD exceeds all comparison attacks in both of transferability and imperceptibility at different comparable levels, demonstrating the effectiveness of the proposed novel modeling framework.

To further elaborate the relationship between transferability and imperceptibility of AdvAD, as well as the optimal trade-off in practice, we plot two line graphs in Figure 4 under more values of $T$. As shown in Figure 4 (a), as the value of $T$ on the horizontal axis changes, the relationship between imperceptibility and transferability shows a clear proportional trend as mentioned above, consistent across different surrogate models. For the optimal trade-off, we consider that the intersection point of the two curves represents a balance between imperceptibility and transferability. Accordingly, for the ResNet-50 and MobileNetV2 models, the optimal values of $T$ are 50 and 25, respectively. Moreover, Figure 4 (b) illustrates more direct curves of this relationship and the positions of other comparison methods within it. Note that, all the other comparison methods are located to the lower left of the curve of AdvAD. This indicates that our method consistently achieves the best results in both transferability and imperceptibility compared with other state-of-the-art restricted imperceptible attacks, demonstrating the effectiveness of our AdvAD as a new attack framework with flexibility through the proposed non-parametric diffusion process.

## 4.5 Analysis

**Eq. (12) in Practice.** With the derived analytical formulation of Proposition 1, in Figure 5, we illustrate the actual values of $\lambda_t$ and $\|\boldsymbol{\delta}_t\|_\infty$ of Eq. (12) using 100 randomly selected images. While Proposition 1 indicates that the upper bound of $\|\boldsymbol{\delta}_t\|_\infty$ is invariant with respect to step $t$, the actual strength of the adversarial guidance produced by AMG rapidly decreases as the process progresses, which validates the unique property given at the end of Sec. 3.2. With the similarly decreasing $\lambda_t$, the whole term of $\lambda_t\|\boldsymbol{\delta}_t\|_\infty$ representing $l_\infty$ distance of the guidance at step $t$ also decreases from about 0.0008 to 0, supporting that the proposed modeling framework performs imperceptible attacks with inherently small perturbation strength. **Performance with Smaller $\xi$.** The results of AdvAD and AdvAD-X with smaller $\xi$ for PC module are shown in Table 5. As $\xi$ decreases from 8 to 2, the imperceptibility is naturally improved because of the upper bound of perturbation becomes lower, yet the ASR of 94.8% only drops slightly. When $\xi = 1/255$, AdvAD still holds 87.4% ASR with 57.87 PSNR and 0.9993 SSIM, which means a large number of examples still can fool the DNN with a maximum of $\pm 1$ modification for each pixel, demonstrating the effectiveness of the adversarial guidance injected in the proposed diffusion process for attacks. Moreover, we provide the ablation study of AdvAD-X and additional discussions in **Appendix** D.3, D.4.

## 5 Conclusion and Outlook

In this paper, we propose a novel, fundamental modeling framework distinct from existing paradigms to tackle the challenge of imperceptible attacks. By exploring and deriving basic theory of diffusion models, the proposed AdvAD performs attacks through a non-parametric diffusion process with adversarial guidance, achieving inherently lower overall perturbation strength with high attack efficacy from a modeling perspective. Besides, the proposed AdvAD-X evaluates the extreme of this novel modeling framework and further reduces the perturbation strength to an extremely low level in an ideal scenario. Extensive experimental results support the effectiveness and progressiveness of the proposed methods. Beyond imperceptibility, AdvAD holds the potential to become a general and extensible attack paradigm thanks to the solid theoretical foundation and the innovative, controllable diffusion-based process for attacks. In addition, we also hope the new observation that AdvAD-X can successfully attack with extremely small perturbation using floating-point raw data can bring inspiration for revealing the robustness and interpretability (e.g., decision boundaries) of DNNs.

## Acknowledgments and Disclosure of Funding

This work was supported by NSFC (Grant No. 62072484), Natural Science Foundation of Guangdong Province (Grant No. 2514050000889) and Guangdong Key Laboratory of Information Security (No. 2023B1212060026).

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

# Contents

# A  Related Work

Beginning with the attack paradigm of Fast Gradient Sign Method (FGSM) [2], there are numerous great works focusing on the imperceptibility of adversarial attacks have been proposed [17–23]. In contrast to another line of attacks aimed at improving the attack success rate and the transferability for black-box models with a more lenient limitation of perturbation strength [13–15], imperceptible adversarial attacks are dedicate to accomplish attacks using as minimal perturbation as possible while deceiving human perception. Among them, PerC-AL [19] improves the imperceptibility by alternating between the classification loss and perceptual color difference when updating perturbations. AdvDrop [21] uses Discrete Cosine Transform (DCT) to discard details in images that are imperceptible for humans. SSAH [24] limits perturbation to high-frequency components using Discrete Wavelet Transform (DWT) to make it undetectable. Similarly, AdvINN [22] also utilizes the DWT and exploits invertible neural networks to specially perform targeted attacks. In addition, with an unrestriced setting [25], some recent works have incorporated the capabilities of generative models (e.g., diffusion models [26]) into common attack frameworks to make adversarial examples more natural and enhance the imperceptibility. DiffAttack [31] and ACA [32] combine the optimization of adversarial losses with the Stable Diffusion [28] to generate unrestricted adversarial examples, while Diff-PGD [30] and AdvDiffuser [29] incorporate the classic PGD method [12] into the diffusion steps to make the adversarial examples undergo denoising processing.

Compared to these traditional restricted imperceptible attacks or the recent unrestricted imperceptible attacks (e.g., diffusion-based), the proposed AdvAD is a completely novel approach distinct from existing attack paradigms. It is the first pilot framework which innovatively conceptualizes attacking as a non-parametric diffusion process by theoretically exploring fundamental modeling approach of diffusion models rather than using their denoising or generative abilities, achieving high attack efficacy and imperceptibility with intrinsically lower perturbation strength. Following the setting of restricted attack, the modeling of AdvAD is theoretically derived from conditional sampling of diffusion models, supporting its attack performance and imperceptibility, and does not require any loss functions, optimizers, or additional neural networks.

# B  Derivations and Proofs

In this section, we first introduce the specific straightforward Pixel-level Constraint (PC) for $\hat{x}_t$ that is simply mentioned at the beginning of Section 3.3 of the main text as an intuitive preliminary, then we provide the detailed proofs of Theorem 1 and Proposition 1, 2 given in the PC for $\hat{\epsilon}_t$.

## B.1  Straightforward PC for $\hat{x}_t$

For each known $\bar{x}_t$ in the fixed diffusion trajectory of the original image $x_{ori}$, and modified $\hat{x}_t$ in the attacking trajectory with the proposed Attacked Model Guidance (AMG) leading to the adversarial example $x_{adv}$, the objective of PC is to control and constrain these two trajectories to be close, ensuring the final $\hat{x}_0$ (i.e., $x_{adv}$) close to $\bar{x}_0$ (i.e., $x_{ori}$). It is obvious that a straightforward way to achieve the goal by directly constrain every $\hat{x}_t$ using $\bar{x}_t$. Thus, in PC for $\hat{x}_t$, we can utilize the restriction of adversarial examples and the relationship between $x_{adv}$, $\hat{x}_t^0$ and $\hat{x}_t$ to derive the constraint for $\hat{x}_t$. Given the budget $\xi$, the desired restriction of adversarial examples is

$$\|x_{adv} - x_{ori}\|_\infty \leq \xi. \tag{15}$$

Next, since the $\hat{\epsilon}_t$ is unconstrained in the case of PC for $\hat{x}_t$, we adopt the initialized $\epsilon_0$ to calculate $\hat{x}_t^0$ approximating $x_{adv}$ as:

$$x_{adv} \approx \hat{x}_t^0(\epsilon_0) = \frac{\hat{x}_t - \sqrt{1 - \alpha_t}\epsilon_0}{\sqrt{\alpha_t}}. \tag{16}$$

where $\alpha_0 = 1$, and $\alpha_{1:T} \in (0, 1]^T$ is a pre-defined decreasing scalar sequence. And the $\bar{x}_t$ of the original trajectory is calculated as:

$$\bar{x}_t = \sqrt{\alpha_t}x_{ori} + \sqrt{1 - \alpha_t}\epsilon_0 \tag{17}$$

By substituting Eq. (16) and Eq. (17) into Eq. (15), the constraint for $\hat{\boldsymbol{x}}_t$ can be easily derived, denoted as:

$$\left\| \frac{\hat{\boldsymbol{x}}_t - \sqrt{1-\alpha_t}\boldsymbol{\epsilon}_0}{\sqrt{\alpha_t}} - \frac{\bar{\boldsymbol{x}}_t - \sqrt{1-\alpha_t}\boldsymbol{\epsilon}_0}{\sqrt{\alpha_t}} \right\|_\infty \leq \xi$$

$$\Leftrightarrow \left\| \frac{\hat{\boldsymbol{x}}_t}{\sqrt{\alpha_t}} - \frac{\bar{\boldsymbol{x}}_t}{\sqrt{\alpha_t}} \right\|_\infty \leq \xi \tag{18}$$

$$\Leftrightarrow \|\hat{\boldsymbol{x}}_t - \bar{\boldsymbol{x}}_t\|_\infty \leq \sqrt{\alpha_t}\,\xi$$

In this way, the PC for $\hat{\boldsymbol{x}}_t$ can be implemented at the start of each step to achieve the basic restrictions by employing a projection operation to $\hat{\boldsymbol{x}}_t$ according to Eq. (18). However, this direct modification of $\hat{\boldsymbol{x}}_t$ at each step obviously disrupts the entire diffusion trajectory from noise to our desired adversarial distribution, and can not achieve the final imperceptibility. Additionally, it is observed that the estimation of $\hat{\boldsymbol{x}}_t^0$ in each step employs a fixed $\boldsymbol{\epsilon}_0$, impairing the adversarial guidance crafted by AMG.

## B.2 Proof of Theorem 1

Therefore, we carefully analysis the important noise term in our diffusion-based modeling approach for adversarial attacks, and present Theorem 1 to support our PC for $\hat{\boldsymbol{\epsilon}}_t$ as described in the main text. The proof of Theorem 1 is provided as follow.

**Theorem 1** *Given diffusion coefficients $\alpha_{T:0} \in (0,1]^T$, the $\boldsymbol{x}_{ori}$, $\bar{\boldsymbol{x}}_t$, $\boldsymbol{\epsilon}_0$ from the original trajectory, $\hat{\boldsymbol{x}}_t$, $\hat{\boldsymbol{\epsilon}}_t$ from the modified trajectory, and a variable $\xi$, if $\hat{\boldsymbol{\epsilon}}_t$ and $\boldsymbol{\epsilon}_0$ satisfies*

$$\|\hat{\boldsymbol{\epsilon}}_t - \boldsymbol{\epsilon}_0\|_\infty \leq \frac{\sqrt{\alpha_T}}{\sqrt{1-\alpha_T}}\xi, \tag{19}$$

*for all $t \in [T:1]$, then it follows that $\|\hat{\boldsymbol{x}}_t - \bar{\boldsymbol{x}}_t\|_\infty \leq (\sqrt{\alpha_t} - \sqrt{1-\alpha_t}\frac{\sqrt{\alpha_T}}{\sqrt{1-\alpha_T}})\xi$, $\|\hat{\boldsymbol{x}}_t^0 - \boldsymbol{x}_{ori}\|_\infty \leq \xi$, and $\|\hat{\boldsymbol{x}}_0 - \boldsymbol{x}_{ori}\|_\infty \leq \xi$ hold true.*

**Proof 1** *We prove Theorem 1 using mathematical induction.*

**Initial case.** *For trajectories of the adversarial example $\boldsymbol{x}_{adv}$ and original image $\boldsymbol{x}_{ori}$ that start from $\hat{\boldsymbol{\epsilon}}_{T+1} = \boldsymbol{\epsilon}_0$, $\hat{\boldsymbol{x}}_T = \bar{\boldsymbol{x}}_T$ and $\hat{\boldsymbol{x}}_T^0 = \bar{\boldsymbol{x}}_T^0$, we can unfold the formula for computing the $\hat{\boldsymbol{x}}_{T-1}^0$ with the updated $\hat{\boldsymbol{\epsilon}}_T$ as:*

$$\begin{aligned}
\hat{\boldsymbol{x}}_{T-1}^0 &= \frac{\hat{\boldsymbol{x}}_{T-1}}{\sqrt{\alpha_{T-1}}} - \frac{\sqrt{1-\alpha_{T-1}}}{\sqrt{\alpha_{T-1}}}\hat{\boldsymbol{\epsilon}}_T \\
&= \frac{\sqrt{\alpha_{T-1}}\left(\frac{\hat{\boldsymbol{x}}_T - \sqrt{1-\alpha_T}\hat{\boldsymbol{\epsilon}}_T}{\sqrt{\alpha_T}}\right) + \sqrt{1-\alpha_{T-1}}\hat{\boldsymbol{\epsilon}}_T}{\sqrt{\alpha_{T-1}}} - \frac{\sqrt{1-\alpha_{T-1}}}{\sqrt{\alpha_{T-1}}}\hat{\boldsymbol{\epsilon}}_T \\
&= \frac{\hat{\boldsymbol{x}}_T - \sqrt{1-\alpha_T}\hat{\boldsymbol{\epsilon}}_T}{\sqrt{\alpha_T}}.
\end{aligned} \tag{20}$$

*For $\bar{\boldsymbol{x}}_{T-1}^0$ from the fixed trajectory where $\bar{\boldsymbol{x}}_t^0$ always equals to $\boldsymbol{x}_{ori}$, we have:*

$$\bar{\boldsymbol{x}}_{T-1}^0 = \frac{\bar{\boldsymbol{x}}_T - \sqrt{1-\alpha_T}\boldsymbol{\epsilon}_0}{\sqrt{\alpha_T}} = \boldsymbol{x}_{ori} \tag{21}$$

*With $\hat{\boldsymbol{x}}_T = \bar{\boldsymbol{x}}_T$, Eq. (20), Eq. (21), and the relationship of $\|\hat{\boldsymbol{\epsilon}}_T - \boldsymbol{\epsilon}_0\|_\infty \leq \frac{\sqrt{\alpha_T}}{\sqrt{1-\alpha_T}}\xi$, we have:*

$$\|\hat{\boldsymbol{\epsilon}}_T - \boldsymbol{\epsilon}_0\|_\infty \leq \frac{\sqrt{\alpha_T}}{\sqrt{1-\alpha_T}}\xi$$

$$\Leftrightarrow \left\| \frac{\sqrt{1-\alpha_T}\hat{\boldsymbol{\epsilon}}_T}{\sqrt{\alpha_T}} - \frac{\sqrt{1-\alpha_T}\boldsymbol{\epsilon}_0}{\sqrt{\alpha_T}} \right\|_\infty \leq \xi$$

$$\Leftrightarrow \left\| \frac{\hat{\boldsymbol{x}}_T - \sqrt{1-\alpha_T}\hat{\boldsymbol{\epsilon}}_T}{\sqrt{\alpha_T}} - \frac{\bar{\boldsymbol{x}}_T - \sqrt{1-\alpha_T}\boldsymbol{\epsilon}_0}{\sqrt{\alpha_T}} \right\|_\infty \leq \xi$$

$$\Leftrightarrow \|\hat{\boldsymbol{x}}_{T-1}^0 - \bar{\boldsymbol{x}}_{T-1}^0\|_\infty \leq \xi \tag{22}$$

Meanwhile, for the relationship between $\hat{\boldsymbol{x}}_{T-1}$ and $\bar{\boldsymbol{x}}_{T-1}$ at the initial step, we have:

$$\|\hat{\boldsymbol{x}}_{T-1} - \bar{\boldsymbol{x}}_{T-1}\|_\infty = \left\| \sqrt{\alpha_{T-1}} \left( \frac{\hat{\boldsymbol{x}}_T - \sqrt{1-\alpha_T}\hat{\boldsymbol{\epsilon}}_T}{\sqrt{\alpha_T}} \right) + \sqrt{1-\alpha_{T-1}}\hat{\boldsymbol{\epsilon}}_T \right.$$

$$\left. -\sqrt{\alpha_{T-1}} \left( \frac{\bar{\boldsymbol{x}}_T - \sqrt{1-\alpha_T}\boldsymbol{\epsilon}_0}{\sqrt{\alpha_T}} \right) - \sqrt{1-\alpha_{T-1}}\boldsymbol{\epsilon}_0 \right\|_\infty$$

$$= \left\| \left( \sqrt{1-\alpha_{T-1}} - \frac{\sqrt{\alpha_{T-1}}\sqrt{1-\alpha_T}}{\sqrt{\alpha_T}} \right) (\hat{\boldsymbol{\epsilon}}_T - \boldsymbol{\epsilon}_0) \right\|_\infty$$

$$= \left| \sqrt{1-\alpha_{T-1}} - \frac{\sqrt{\alpha_{T-1}}\sqrt{1-\alpha_T}}{\sqrt{\alpha_T}} \right| \|\hat{\boldsymbol{\epsilon}}_T - \boldsymbol{\epsilon}_0\|_\infty \tag{23}$$

$$= \left( \frac{\sqrt{\alpha_{T-1}}\sqrt{1-\alpha_T}}{\sqrt{\alpha_T}} - \sqrt{1-\alpha_{T-1}} \right) \|\hat{\boldsymbol{\epsilon}}_T - \boldsymbol{\epsilon}_0\|_\infty \tag{24}$$

$$\leq \left( \sqrt{\alpha_{T-1}} - \sqrt{1-\alpha_{T-1}}\frac{\sqrt{\alpha_T}}{\sqrt{1-\alpha_T}} \right) \xi, \tag{25}$$

where the transition from Eq. (23) to Eq. (24) is obtained with the real constant value of $\alpha_t$. At this point, it can be seen that Theorem 1 holds in the initial case.

**Inductive step.** *Assuming the theorem holds for some arbitrary step $k$, where $T \geq k > 1$, we have:*

$$\|\hat{\boldsymbol{\epsilon}}_{k+1} - \boldsymbol{\epsilon}_0\|_\infty \leq \frac{\sqrt{\alpha_T}}{\sqrt{1-\alpha_T}}\xi, \tag{26}$$

$$\|\hat{\boldsymbol{x}}_k^0 - \boldsymbol{x}_{ori}\|_\infty = \|\hat{\boldsymbol{x}}_k^0 - \bar{\boldsymbol{x}}_k^0\|_\infty \leq \xi, \tag{27}$$

*and*

$$\|\hat{\boldsymbol{x}}_k - \bar{\boldsymbol{x}}_k\|_\infty \leq (\sqrt{\alpha_k} - \sqrt{1-\alpha_k}\frac{\sqrt{\alpha_T}}{\sqrt{1-\alpha_T}})\xi. \tag{28}$$

*Based on the inductive hypothesis, we next show the validity of the theorem at step $k-1$. Similar to Eq.(20), we unfold the calculation of $\hat{\boldsymbol{x}}_{k-1}^0$ with $\hat{\boldsymbol{x}}_k$ and $\hat{\boldsymbol{\epsilon}}_k$ as:*

$$\hat{\boldsymbol{x}}_{k-1}^0 = \frac{\hat{\boldsymbol{x}}_{k-1}}{\sqrt{\alpha_{k-1}}} - \frac{\sqrt{1-\alpha_{k-1}}}{\sqrt{\alpha_{k-1}}}\hat{\boldsymbol{\epsilon}}_k$$

$$= \frac{\sqrt{\alpha_{k-1}} \left( \frac{\hat{\boldsymbol{x}}_k - \sqrt{1-\alpha_k}\hat{\boldsymbol{\epsilon}}_k}{\sqrt{\alpha_k}} \right) + \sqrt{1-\alpha_{k-1}}\hat{\boldsymbol{\epsilon}}_k}{\sqrt{\alpha_{k-1}}} - \frac{\sqrt{1-\alpha_{k-1}}}{\sqrt{\alpha_{k-1}}}\hat{\boldsymbol{\epsilon}}_k$$

$$= \frac{\hat{\boldsymbol{x}}_k - \sqrt{1-\alpha_k}\hat{\boldsymbol{\epsilon}}_k}{\sqrt{\alpha_k}}. \tag{29}$$

*And the $\bar{\boldsymbol{x}}_{k-1}^0$ can be denoted as:*

$$\bar{\boldsymbol{x}}_{k-1}^0 = \frac{\bar{\boldsymbol{x}}_k - \sqrt{1-\alpha_k}\boldsymbol{\epsilon}_0}{\sqrt{\alpha_k}} = \boldsymbol{x}_{ori} \tag{30}$$

*Consequently, by substituting Eq. (29) and Eq. (30) into $\left\|\hat{\boldsymbol{x}}_{k-1}^0 - \bar{\boldsymbol{x}}_{k-1}^0\right\|_\infty$, we have:*

$$\left\|\hat{\boldsymbol{x}}_{k-1}^0 - \bar{\boldsymbol{x}}_{k-1}^0\right\|_\infty = \left\| \frac{\hat{\boldsymbol{x}}_k - \sqrt{1-\alpha_k}\hat{\boldsymbol{\epsilon}}_k}{\sqrt{\alpha_k}} - \frac{\bar{\boldsymbol{x}}_k - \sqrt{1-\alpha_k}\boldsymbol{\epsilon}_0}{\sqrt{\alpha_k}} \right\|_\infty$$

$$= \left\| \frac{1}{\sqrt{\alpha_k}} (\hat{\boldsymbol{x}}_k - \bar{\boldsymbol{x}}_k) + \frac{\sqrt{1-\alpha_k}}{\sqrt{\alpha_k}} (\boldsymbol{\epsilon}_0 - \hat{\boldsymbol{\epsilon}}_k) \right\|_\infty \tag{31}$$

$$\leq \frac{1}{\sqrt{\alpha_k}} \|\hat{\boldsymbol{x}}_k - \bar{\boldsymbol{x}}_k\|_\infty + \frac{\sqrt{1-\alpha_k}}{\sqrt{\alpha_k}} \|\hat{\boldsymbol{\epsilon}}_k - \boldsymbol{\epsilon}_0\|_\infty \tag{32}$$

$$\leq \left( 1 - \frac{\sqrt{1-\alpha_k}}{\sqrt{\alpha_k}} \frac{\sqrt{\alpha_T}}{\sqrt{1-\alpha_T}} \right) \xi + \frac{\sqrt{1-\alpha_k}}{\sqrt{\alpha_k}} \|\hat{\boldsymbol{\epsilon}}_k - \boldsymbol{\epsilon}_0\|_\infty , \tag{33}$$

*where Eq. (31) to Eq. (32) utilizes the triangle inequality property of $l_p$-norm, and Eq. (33) is obtained with Eq. (28). Then, given the imposed condition of Eq. (19), we can get:*

$$\|\hat{\boldsymbol{\epsilon}}_k - \boldsymbol{\epsilon}_0\|_\infty \leq \frac{\sqrt{\alpha_T}}{\sqrt{1-\alpha_T}}\xi$$

$$\Leftrightarrow \left(1 - \frac{\sqrt{1-\alpha_k}}{\sqrt{\alpha_k}}\frac{\sqrt{\alpha_T}}{\sqrt{1-\alpha_T}}\right)\xi + \frac{\sqrt{1-\alpha_k}}{\sqrt{\alpha_k}}\|\hat{\boldsymbol{\epsilon}}_k - \boldsymbol{\epsilon}_0\|_\infty \leq \xi$$

$$\Leftrightarrow \left\|\hat{\boldsymbol{x}}_{k-1}^0 - \bar{\boldsymbol{x}}_{k-1}^0\right\|_\infty \leq \xi$$

$$\Leftrightarrow \left\|\hat{\boldsymbol{x}}_{k-1}^0 - \boldsymbol{x}_{ori}\right\|_\infty \leq \xi, \tag{34}$$

*And the relationship between $\hat{\boldsymbol{x}}_{k-1}$ and $\bar{\boldsymbol{x}}_{k-1}$ at step $k-1$ can be expressed as:*

$$\|\hat{\boldsymbol{x}}_{k-1} - \bar{\boldsymbol{x}}_{k-1}\|_\infty$$
$$= \left\|\sqrt{\alpha_{k-1}}\left(\frac{\hat{\boldsymbol{x}}_k - \sqrt{1-\alpha_k}\hat{\boldsymbol{\epsilon}}_k}{\sqrt{\alpha_k}}\right) + \sqrt{1-\alpha_{k-1}}\hat{\boldsymbol{\epsilon}}_k \right.$$
$$\left. - \sqrt{\alpha_{k-1}}\left(\frac{\bar{\boldsymbol{x}}_k - \sqrt{1-\alpha_k}\boldsymbol{\epsilon}_0}{\sqrt{\alpha_k}}\right) - \sqrt{1-\alpha_{k-1}}\boldsymbol{\epsilon}_0\right\|_\infty$$

$$= \left\|\frac{\sqrt{\alpha_{k-1}}}{\sqrt{\alpha_k}}(\hat{\boldsymbol{x}}_k - \bar{\boldsymbol{x}}_k) + \left(\sqrt{1-\alpha_{k-1}} - \frac{\sqrt{\alpha_{k-1}}\sqrt{1-\alpha_k}}{\sqrt{\alpha_k}}\right)(\hat{\boldsymbol{\epsilon}}_k - \boldsymbol{\epsilon}_0)\right\|_\infty \tag{35}$$

$$\leq \frac{\sqrt{\alpha_{k-1}}}{\sqrt{\alpha_k}}\|\hat{\boldsymbol{x}}_k - \bar{\boldsymbol{x}}_k\|_\infty + \left|\sqrt{1-\alpha_{k-1}} - \frac{\sqrt{\alpha_{k-1}}\sqrt{1-\alpha_k}}{\sqrt{\alpha_k}}\right|\|\hat{\boldsymbol{\epsilon}}_k - \boldsymbol{\epsilon}_0\|_\infty \tag{36}$$

$$\leq \frac{\sqrt{\alpha_{k-1}}}{\sqrt{\alpha_k}}\left(\sqrt{\alpha_k} - \sqrt{1-\alpha_k}\frac{\sqrt{\alpha_T}}{\sqrt{1-\alpha_T}}\right)\xi + \left(\frac{\sqrt{\alpha_{k-1}}\sqrt{1-\alpha_k}}{\sqrt{\alpha_k}} - \sqrt{1-\alpha_{k-1}}\right)\frac{\sqrt{\alpha_T}}{\sqrt{1-\alpha_T}}\xi \tag{37}$$

$$= \left(\sqrt{\alpha_{k-1}} - \sqrt{1-\alpha_{k-1}}\frac{\sqrt{\alpha_T}}{\sqrt{1-\alpha_T}}\right)\xi \tag{38}$$

*where the triangle inequality property is utilized again to obtain Eq. (36) from Eq. (35), then Eq. (28) and Eq. (34) is substituted to obtain Eq. (37). Obviously, for the case of step $k-1$, it is also consistent with the theorem.*

**Conclusion.** *Therefore, by extending the truth of the theorem from arbitrary step $k$ to $k-1$, and given its established validity at the initial case, the principle of mathematical induction allows us to conclude that $\|\hat{\boldsymbol{x}}_t - \bar{\boldsymbol{x}}_t\|_\infty \leq (\sqrt{\alpha_t} - \sqrt{1-\alpha_t}\frac{\sqrt{\alpha_T}}{\sqrt{1-\alpha_T}})\xi$, $\|\hat{\boldsymbol{x}}_t^0 - \boldsymbol{x}_{ori}\|_\infty \leq \xi$ hold true for every step $t \in [1:T]$. For $t=0$ and $\alpha_0 = 1$, we have $\hat{\boldsymbol{x}}_0^0 = \hat{\boldsymbol{x}}_0$ and $\bar{\boldsymbol{x}}_0 = \boldsymbol{x}_{ori}$, thus it is obvious that $\|\hat{\boldsymbol{x}}_0 - \boldsymbol{x}_{ori}\|_\infty \leq \xi$. This concludes the proof of the whole Theorem 1.*

### B.3 Proof of Proposition 1

Next, we prove Proposition 1 about $\lambda_t$ and $\boldsymbol{\delta}_t$ by expanding and rearranging the recursive formulas in our diffusion process for attacks.

**Proposition 1** *Under the conditions of Theorem 1, by denoting constrained $\hat{\boldsymbol{\epsilon}}_t = \boldsymbol{\epsilon}_0 - \boldsymbol{\delta}_t$, we have*

$$\boldsymbol{x}_{adv} = \boldsymbol{x}_{ori} + \sum_{t=1}^{T}\lambda_t\boldsymbol{\delta}_t, \tag{39}$$

*where $\lambda_t = \frac{\sqrt{1-\alpha_t}}{\sqrt{\alpha_t}} - \frac{\sqrt{1-\alpha_{t-1}}}{\sqrt{\alpha_{t-1}}}$, and $\|\boldsymbol{\delta}_t\|_\infty \leq \frac{\sqrt{\alpha_T}}{\sqrt{1-\alpha_T}}\xi$.*

**Proof 2** *With Eq.* (19) *in Theorem 1, by denoting* $\hat{\epsilon}_t = \epsilon_0 - \delta_t$, *we have:* $\|\delta_t\|_\infty = \|\hat{\epsilon}_t - \epsilon_0\|_\infty \leq \frac{\sqrt{\alpha_T}}{\sqrt{1-\alpha_T}}\xi$, *and the* $\hat{x}_{T-1}$ *in the initial step can be written as:*

$$
\begin{aligned}
\hat{x}_{T-1} &= \frac{\sqrt{\alpha_{T-1}}}{\sqrt{\alpha_T}}\hat{x}_T - \left(\frac{\sqrt{\alpha_{T-1}}\sqrt{1-\alpha_T}}{\sqrt{\alpha_T}} - \sqrt{1-\alpha_{T-1}}\right)(\hat{\epsilon}_T) \\
&= \frac{\sqrt{\alpha_{T-1}}}{\sqrt{\alpha_T}}\hat{x}_T - \sqrt{\alpha_{T-1}}\left(\frac{\sqrt{1-\alpha_T}}{\sqrt{\alpha_T}} - \frac{\sqrt{1-\alpha_{T-1}}}{\sqrt{\alpha_{T-1}}}\right)(\epsilon_0 - \delta_T)
\end{aligned}
\tag{40}
$$

*Applying the recursion formula twice, we have:*

$$
\begin{aligned}
\hat{x}_{T-2} &= \frac{\sqrt{\alpha_{T-2}}}{\sqrt{\alpha_{T-1}}}\left(\frac{\sqrt{\alpha_{T-1}}}{\sqrt{\alpha_T}}\hat{x}_T - \left(\frac{\sqrt{\alpha_{T-1}}\sqrt{1-\alpha_T}}{\sqrt{\alpha_T}} - \sqrt{1-\alpha_{T-1}}\right)(\hat{\epsilon}_T)\right) \\
&\quad - \left(\frac{\sqrt{\alpha_{T-2}}\sqrt{1-\alpha_{T-1}}}{\sqrt{\alpha_{T-1}}} - \sqrt{1-\alpha_{T-2}}\right)(\hat{\epsilon}_{T-1}) \\[2mm]
&= \frac{\sqrt{\alpha_{T-2}}}{\sqrt{\alpha_T}}\hat{x}_T - \sqrt{\alpha_{T-2}}\left(\frac{\sqrt{1-\alpha_T}}{\sqrt{\alpha_T}} - \frac{\sqrt{1-\alpha_{T-1}}}{\sqrt{\alpha_{T-1}}}\right)(\epsilon_0 - \delta_T) \\
&\quad - \sqrt{\alpha_{T-2}}\left(\frac{\sqrt{1-\alpha_{T-1}}}{\sqrt{\alpha_{T-1}}} - \frac{\sqrt{1-\alpha_{T-2}}}{\sqrt{\alpha_{T-2}}}\right)(\epsilon_0 - \delta_{T-1})
\end{aligned}
\tag{41}
$$

*Similarly, for* $\hat{x}_{T-3}$, *we have:*

$$
\begin{aligned}
\hat{x}_{T-3} &= \frac{\sqrt{\alpha_{T-3}}}{\sqrt{\alpha_{T-2}}}\left(\frac{\sqrt{\alpha_{T-2}}}{\sqrt{\alpha_{T-1}}}\left(\frac{\sqrt{\alpha_{T-1}}}{\sqrt{\alpha_T}}\hat{x}_T - \left(\frac{\sqrt{\alpha_{T-1}}\sqrt{1-\alpha_T}}{\sqrt{\alpha_T}} - \sqrt{1-\alpha_{T-1}}\right)(\hat{\epsilon}_T)\right)\right. \\
&\quad\quad - \left(\frac{\sqrt{\alpha_{T-2}}\sqrt{1-\alpha_{T-1}}}{\sqrt{\alpha_{T-1}}} - \sqrt{1-\alpha_{T-2}}\right)(\hat{\epsilon}_{T-1})\Bigg) \\
&\quad - \left(\frac{\sqrt{\alpha_{T-3}}\sqrt{1-\alpha_{T-2}}}{\sqrt{\alpha_{T-2}}} - \sqrt{1-\alpha_{T-3}}\right)(\hat{\epsilon}_{T-2}) \\[2mm]
&= \frac{\sqrt{\alpha_{T-3}}}{\sqrt{\alpha_T}}\hat{x}_T - \sqrt{\alpha_{T-3}}\left(\frac{\sqrt{1-\alpha_T}}{\sqrt{\alpha_T}} - \frac{\sqrt{1-\alpha_{T-1}}}{\sqrt{\alpha_{T-1}}}\right)(\epsilon_0 - \delta_T) \\
&\quad - \sqrt{\alpha_{T-3}}\left(\frac{\sqrt{1-\alpha_{T-1}}}{\sqrt{\alpha_{T-1}}} - \frac{\sqrt{1-\alpha_{T-2}}}{\sqrt{\alpha_{T-2}}}\right)(\epsilon_0 - \delta_{T-1}) \\
&\quad - \sqrt{\alpha_{T-3}}\left(\frac{\sqrt{1-\alpha_{T-2}}}{\sqrt{\alpha_{T-2}}} - \frac{\sqrt{1-\alpha_{T-3}}}{\sqrt{\alpha_{T-3}}}\right)(\epsilon_0 - \delta_{T-2})
\end{aligned}
\tag{42}
$$

*It is obvious that the coefficients of each term exhibit a clear regular pattern related to the step* $t$. *Following this pattern, we can accordingly get the expression of* $\hat{x}_t$ *as:*

$$
\begin{aligned}
\hat{x}_t &= \frac{\sqrt{\alpha_t}}{\sqrt{\alpha_T}}\hat{x}_T - \sqrt{\alpha_t}\left(\frac{\sqrt{1-\alpha_T}}{\sqrt{\alpha_T}} - \frac{\sqrt{1-\alpha_{T-1}}}{\sqrt{\alpha_{T-1}}}\right)(\epsilon_0 - \delta_T) \\
&\quad - \sqrt{\alpha_t}\left(\frac{\sqrt{1-\alpha_{T-1}}}{\sqrt{\alpha_{T-1}}} - \frac{\sqrt{1-\alpha_{T-2}}}{\sqrt{\alpha_{T-2}}}\right)(\epsilon_0 - \delta_{T-1}) \\
&\quad \vdots \\
&\quad - \sqrt{\alpha_t}\left(\frac{\sqrt{1-\alpha_{t+2}}}{\sqrt{\alpha_{t+2}}} - \frac{\sqrt{1-\alpha_{t+1}}}{\sqrt{\alpha_{t+1}}}\right)(\epsilon_0 - \delta_{t+2}) \\
&\quad - \sqrt{\alpha_t}\left(\frac{\sqrt{1-\alpha_{t+1}}}{\sqrt{\alpha_{t+1}}} - \frac{\sqrt{1-\alpha_t}}{\sqrt{\alpha_t}}\right)(\epsilon_0 - \delta_{t+1})
\end{aligned}
\tag{43}
$$

And the final $\hat{x}_0$ can be expressed as:

$$\hat{x}_0 = \frac{\sqrt{\alpha_0}}{\sqrt{\alpha_T}}\hat{x}_T - \sqrt{\alpha_0}\left(\frac{\sqrt{1-\alpha_T}}{\sqrt{\alpha_T}} - \frac{\sqrt{1-\alpha_{T-1}}}{\sqrt{\alpha_{T-1}}}\right)(\epsilon_0 - \delta_T)$$
$$- \sqrt{\alpha_0}\left(\frac{\sqrt{1-\alpha_{T-1}}}{\sqrt{\alpha_{T-1}}} - \frac{\sqrt{1-\alpha_{T-2}}}{\sqrt{\alpha_{T-2}}}\right)(\epsilon_0 - \delta_{T-1})$$
$$\vdots$$
$$- \sqrt{\alpha_0}\left(\frac{\sqrt{1-\alpha_2}}{\sqrt{\alpha_2}} - \frac{\sqrt{1-\alpha_1}}{\sqrt{\alpha_1}}\right)(\epsilon_0 - \delta_2)$$
$$- \sqrt{\alpha_0}\left(\frac{\sqrt{1-\alpha_1}}{\sqrt{\alpha_1}} - \frac{\sqrt{1-\alpha_0}}{\sqrt{\alpha_0}}\right)(\epsilon_0 - \delta_1) \tag{44}$$

Note that in $\alpha_0 = 1$ Eq. (44), and the coefficients of $\epsilon_0$ can be mostly eliminated. Thus, by defining $\lambda_t$ as:

$$\lambda_t = \frac{\sqrt{1-\alpha_t}}{\sqrt{\alpha_t}} - \frac{\sqrt{1-\alpha_{t-1}}}{\sqrt{\alpha_{t-1}}}, \tag{45}$$

we can rearrange Eq. (44) into:

$$\hat{x}_0 = \frac{\hat{x}_T - \sqrt{1-\alpha_T}\epsilon_0}{\sqrt{\alpha_T}} + \sum_{t=1}^{T}\lambda_t\delta_t$$
$$= \frac{\bar{x}_T - \sqrt{1-\alpha_T}\epsilon_0}{\sqrt{\alpha_T}} + \sum_{t=1}^{T}\lambda_t\delta_t$$
$$= x_{ori} + \sum_{t=1}^{T}\lambda_t\delta_t \tag{46}$$

where $\hat{x}_0$ is the final output of the diffusing process and $x_{adv} = \hat{x}_0$. This concludes the proof of Proposition 1.

### B.4    Proof of Proposition 2

Finally, we prove Proposition 2 about the validity and convergence of the approximation $x_{adv} \approx \hat{x}_t^0$.

**Proposition 2** *Under the conditions of Theorem 1 and Proposition 1, the upper bound on the error of the approximation in Eq. (22) can be expressed as*

$$\left\|x_{adv} - \hat{x}_t^0\right\|_\infty \le 2 \cdot \frac{\sqrt{1-\alpha_t}}{\sqrt{\alpha_t}} \cdot \frac{\sqrt{\alpha_T}}{\sqrt{1-\alpha_T}}. \tag{47}$$

**Proof 3** *With Eq. (43) and the definitions of Proposition 1, $\hat{x}_t$ and $\hat{x}_t^0$ can be written as:*

$$\hat{x}_t = \frac{\sqrt{\alpha_t}}{\sqrt{\alpha_T}}\hat{x}_T - \sqrt{\alpha_t}\sum_{k=t+1}^{T}\lambda_k(\epsilon_0 - \delta_k), \tag{48}$$

*and*

$$\hat{x}_t^0 = \frac{\hat{x}_t - \sqrt{1-\alpha_t}\hat{\epsilon}_{t+1}}{\sqrt{\alpha_t}} = \frac{1}{\sqrt{\alpha_T}}\hat{x}_T - \sum_{k=t+1}^{T}\lambda_k(\epsilon_0 - \delta_k) - \frac{\sqrt{1-\alpha_t}}{\sqrt{\alpha_t}}(\epsilon_0 - \delta_{t+1}). \tag{49}$$

*For $x_{adv}$, we have:*

$$x_{adv} = \hat{x}_0 = \frac{\hat{x}_T - \sqrt{1-\alpha_T}\epsilon_0}{\sqrt{\alpha_T}} + \sum_{t=1}^{T}\lambda_t\delta_t \tag{50}$$

*With Eq. (49) and Eq. (50), we can obtain:*

$$x_{adv} - \hat{x}_t^0 = \sum_{k=1}^{t}\lambda_k\delta_k - \frac{\sqrt{1-\alpha_t}}{\sqrt{\alpha_t}}\delta_{t+1} \tag{51}$$

*Thus, we have:*

$$
\begin{aligned}
\left\| \boldsymbol{x}_{adv} - \hat{\boldsymbol{x}}_t^0 \right\|_\infty &= \left\| \sum_{k=1}^{t} \lambda_k \boldsymbol{\delta}_k - \frac{\sqrt{1-\alpha_t}}{\sqrt{\alpha_t}} \boldsymbol{\delta}_{t+1} \right\|_\infty \\
&\leq \left\| \sum_{k=1}^{t} \lambda_k \boldsymbol{\delta}_k \right\|_\infty + \left\| \frac{\sqrt{1-\alpha_t}}{\sqrt{\alpha_t}} \boldsymbol{\delta}_{t+1} \right\|_\infty \\
&= \sum_{k=1}^{t} \lambda_k \left\| \boldsymbol{\delta}_k \right\|_\infty + \frac{\sqrt{1-\alpha_t}}{\sqrt{\alpha_t}} \left\| \boldsymbol{\delta}_{t+1} \right\|_\infty \\
&\leq \sum_{k=1}^{t} \left( \lambda_k \cdot \frac{\sqrt{\alpha_T}}{\sqrt{1-\alpha_T}} \xi \right) + \frac{\sqrt{1-\alpha_t}}{\sqrt{\alpha_t}} \frac{\sqrt{\alpha_T}}{\sqrt{1-\alpha_T}} \xi \\
&= \left( \sum_{k=1}^{t} \lambda_k \right) \cdot \frac{\sqrt{\alpha_T}}{\sqrt{1-\alpha_T}} \xi + \frac{\sqrt{1-\alpha_t}}{\sqrt{\alpha_t}} \frac{\sqrt{\alpha_T}}{\sqrt{1-\alpha_T}} \xi \\
&= \left( \frac{\sqrt{1-\alpha_t}}{\sqrt{\alpha_t}} - \frac{\sqrt{1-\alpha_0}}{\sqrt{\alpha_0}} \right) \cdot \frac{\sqrt{\alpha_T}}{\sqrt{1-\alpha_T}} \xi + \frac{\sqrt{1-\alpha_t}}{\sqrt{\alpha_t}} \frac{\sqrt{\alpha_T}}{\sqrt{1-\alpha_T}} \xi \\
&= 2 \cdot \frac{\sqrt{1-\alpha_t}}{\sqrt{\alpha_t}} \frac{\sqrt{\alpha_T}}{\sqrt{1-\alpha_T}} \xi
\end{aligned}
\tag{52}
$$

*This concludes the proof of Proposition 2.*

## C  Algorithm of AdvAD-X

---
**Algorithm 2 AdvAD-X**

---
**Input**: Attacked model $f(\cdot)$, image $\boldsymbol{x}_{ori}$ with label $y_{gt}$, budget $\xi$, step $T$;
**Output**: Adversarial example $\boldsymbol{x}_{adv}$

1: Initialize pre-defined diffusion coefficients $\alpha_{0:T} \in (0,1]^{T+1}$;
2: Initialize $\boldsymbol{\epsilon}_0 \sim \mathcal{N}(\boldsymbol{0}, \boldsymbol{I})$;                    ▷ Initialize and fix diffusion noise $\boldsymbol{\epsilon}_0$.
3: Transform the range of $\boldsymbol{x}_{ori}$ to [-1, 1];                ▷ Align with data range of diffusion process.
4: Calculate $\bar{\boldsymbol{x}}_T$ via Eq. (4);                      ▷ Forward process of adding noise $\boldsymbol{\epsilon}_0$ to $\boldsymbol{x}_{ori}$.
5: Set $\hat{\boldsymbol{x}}_T := \bar{\boldsymbol{x}}_T$, $\hat{\boldsymbol{\epsilon}}_{T+1} := \boldsymbol{\epsilon}_0$;                    ▷ Non-parametric diffusion process.
6: Calculate mask $\boldsymbol{m}$ of $\boldsymbol{x}_{ori}$ using GradCAM;                ▷ Mask $\boldsymbol{m}$ for the CA strategy.
7: **for** $t = T$ to 1 **do**
8:     Calculate $\hat{\boldsymbol{x}}_t^0$ via Eq. (7);                              ▷ Approximation of $\hat{\boldsymbol{x}}_t^0 \approx \boldsymbol{x}_{adv}$.
9:     Transform the range of $\hat{\boldsymbol{x}}_t^0$ to [0, 255];            ▷ Align with data range of image.
10:    *// DGI strategy for performing AMG and PC dynamically.*
11:    **if** $f(\hat{\boldsymbol{x}}_t^0) == y_{gt}$ **then**
12:        Calculate $\hat{\boldsymbol{\epsilon}}_t'$ using $\boldsymbol{m}$ with AMG via Eq. (14);        ▷ AMG module and CA strategy.
13:        Calculate $\hat{\boldsymbol{\epsilon}}_t$ with PC via Eq. (10);              ▷ Same PC module as AdvAD.
14:    **else**
15:        Set $\hat{\boldsymbol{\epsilon}}_t = \boldsymbol{\epsilon}_0$;                          ▷ Skip the operations of current step.
16:    Calculate $\hat{\boldsymbol{x}}_{t-1}$ via Eq. (11);              ▷ One step backward from $t$ to $t-1$.
17: Transform the range of $\hat{\boldsymbol{x}}_0$ to [0, 255];                    ▷ Endpoint of the process.
18: **return** $\boldsymbol{x}_{adv} = \hat{\boldsymbol{x}}_0$;      ▷ Directly return $\boldsymbol{x}_{adv}$ in raw floating-point data for the ideal scenario.

---

## D  Additional Experiments

### D.1  Additional Quantitative Comparisons

Table 6 reports the untargeted attack performance and imperceptibility of ten methods on Vgg-19, MobileNet-V2, and WideResNet-50 models. The results indicate that the proposed AdvAD and AdvAD-X, leveraging a novel modeling framework, consistently achieve superior performance. These findings further underscore the effectiveness of the proposed approach.

Table 6: Additional results of untargeted white-box attack success rate (ASR) and other evaluation metrics for imperceptibility when employing different attacks and attacked models. The reported running times are obtained using a RTX 3090 GPU on a same machine. † and blue mean the results of AdvAD-X are obtained with floating-point data type in the ideal scenario as described in Sec 3.4.

| Model | Attack Method | Time (s) ↓ | ASR (%) ↑ | $l_\infty$ ↓ | $l_2$ ↓ | PSNR ↑ | SSIM ↑ | FID ↓ | LPIPS ↓ | MUSIQ ↑ |
|---|---|---|---|---|---|---|---|---|---|---|
| Vgg-19 | PGD | 47 | **100.0** | 0.031 | 8.47 | 33.23 | 0.8771 | 43.15 | 0.0508 | 53.51 |
| | NCF | 3288 | 92.8 | 0.794 | 75.21 | 14.77 | 0.6391 | 57.45 | 0.3077 | 49.27 |
| | ACA | 83123 | 93.4 | 0.832 | 51.22 | 18.22 | 0.5767 | 66.78 | 0.3277 | 55.61 |
| | DiffAttack | 34163 | 97.0 | 0.769 | 31.92 | 22.23 | 0.6632 | 59.08 | 0.1235 | **57.22** |
| | DiffPGD | 5770 | 93.9 | 0.246 | 11.46 | 30.93 | 0.8888 | 20.72 | 0.0317 | 55.23 |
| | AdvDrop | 268 | 97.5 | 0.062 | 3.23 | 41.79 | 0.9867 | 5.90 | 0.0061 | 54.93 |
| | PerC-AL | 8671 | **100.0** | 0.142 | 2.12 | 45.92 | 0.9885 | 10.78 | 0.0028 | 55.91 |
| | SSAH | 948 | 85.5 | 0.027 | 2.35 | 44.62 | 0.9920 | 4.25 | 0.0017 | 55.45 |
| | AdvAD (ours) | 4370 | 99.5 | **0.009** | **1.05** | **52.13** | **0.9979** | **2.62** | **0.0005** | 56.31 |
| | AdvAD-X†(ours) | 1967 | **99.9** | **0.001** | **0.32** | **64.76** | **0.9997** | **0.27** | **0.0001** | **56.56** |
| MobileNet-V2 | PGD | 10 | 99.9 | 0.031 | 8.29 | 33.41 | 0.8803 | 34.57 | 0.0500 | 52.00 |
| | NCF | 2503 | 92.5 | 0.784 | 76.02 | 14.69 | 0.6373 | 56.23 | 0.3090 | 49.37 |
| | ACA | 83118 | 92.8 | 0.835 | 50.70 | 18.30 | 0.5786 | 64.90 | 0.3254 | 56.17 |
| | DiffAttack | 34723 | 98.2 | 0.739 | 30.51 | 22.61 | 0.6733 | 55.77 | 0.1143 | 56.01 |
| | DiffPGD | 5941 | 92.6 | 0.246 | 11.43 | 30.95 | 0.8887 | 19.22 | 0.0309 | 54.87 |
| | AdvDrop | 116 | 97.7 | 0.063 | 3.16 | 41.94 | 0.9873 | 4.88 | 0.0064 | 54.91 |
| | PerC-AL | 3187 | 99.8 | 0.118 | 2.16 | 45.67 | 0.9879 | 8.77 | 0.0032 | 55.59 |
| | SSAH | 265 | 97.8 | 0.026 | 2.18 | 45.24 | 0.9930 | 2.94 | 0.0016 | 55.78 |
| | AdvAD (ours) | 992 | 99.6 | **0.008** | **0.94** | **53.07** | **0.9982** | **1.46** | **0.0004** | **56.37** |
| | AdvAD-X†(ours) | 388 | **100.0** | **0.001** | **0.24** | **66.8** | **0.9998** | **0.11** | **0.0001** | **56.59** |
| WideResNet-50 | PGD | 42 | 96.0 | 0.031 | 8.2 | 33.5 | 0.8830 | 35.594 | 0.0521 | 52.43 |
| | NCF | 2971 | 89.7 | 0.777 | 74.05 | 14.98 | 0.6473 | 56.01 | 0.2965 | 49.45 |
| | ACA | 84163 | 88.0 | 0.838 | 53.17 | 17.89 | 0.5619 | 68.27 | 0.3442 | 55.47 |
| | DiffAttack | 34072 | 95.1 | 0.747 | 30.61 | 22.60 | 0.6737 | 54.71 | 0.1137 | 55.68 |
| | DiffPGD | 5965 | 91.4 | 0.245 | 11.44 | 30.95 | 0.8905 | 21.24 | 0.0317 | 55.16 |
| | AdvDrop | 353 | 96.5 | 0.062 | 3.28 | 41.64 | 0.9863 | 6.21 | 0.0060 | 54.917 |
| | PerC-AL | 6655 | 97.8 | 0.133 | 1.91 | 46.80 | 0.9906 | 9.28 | 0.0025 | 56.07 |
| | SSAH | 738 | 95.7 | 0.028 | 2.21 | 45.21 | 0.9933 | 3.95 | 0.0015 | 55.88 |
| | AdvAD (ours) | 3845 | **99.9** | **0.010** | **1.10** | **51.54** | **0.9979** | **2.84** | **0.0006** | **56.33** |
| | AdvAD-X†(ours) | 1477 | **100.0** | **0.002** | **0.38** | **62.54** | **0.9996** | **0.33** | **0.0001** | **56.58** |

## D.2 Additional Visualizations

More visualizations of adversarial examples and perturbations under the attacked model of ResNet-50 are displayed in Figure 6. The visualizations provide a clear insight into how different methods accomplish imperceptible attacks. Our AdvAD and AdvAD-X methods execute imperceptible attacks with lower overall intensity of perturbations. Notably, for images with salient objects (e.g., the bridge in the second examples), the perturbation intensity of AdvAD also naturally increases in the object regions during the gradient-based adversarial guidance calculation, while AdvAD-X, equipped with the dynamic strategy, still shows uniform and lower overall perturbation intensity.

## D.3 Ablation Study of AdvAD-X

From AdvAD to AdvAD-X, Table 7 shows the effect of the two strategies of CA and DGI. It can be observed that adding CA in each step of AdvAD slightly improves imprecepbility while maintaining the attack success rate of 100%. However, the DGI strategy significantly reduces the iterations of performing AMG and PC from 1000 to 3.97, which indicates that our framework theoretically only requires very little injected adversarial guidance to successfully perform attacks, proving the performance of our modeling method as well as the effectiveness of the adversarial guidance. In AdvAD-X, which finally uses both CA and DGI, the guidance strength in each step is further suppressed, resulting in a slight increase in the total number of iterations required adaptively, but the final perturbation strength continues to decrease to a more extreme level.

Table 7: Ablation study of the proposed CAM Assistance (CA) and Dynamic Gradient Injection (DGI) strategies in AdvAD-X. As marked with †, all the results in this experiment are obtained with attacking normal ResNet-50 using the floating-point raw data to align with the setting of AdvAD-X. The term of Iter. indicates the number of iterations that the AMG and PC are performed.

| Attack | Iter. | ASR↑ | $l_2$↓ | PSNR↑ | SSIM↑ | FID↓ |
|---|---|---|---|---|---|---|
| AdvAD† | 1000 | **100.0** | 0.97 | 52.60 | 0.9984 | 2.3894 |
| AdvAD+CA† | 1000 | **100.0** | 0.89 | 53.27 | 0.9987 | 2.2033 |
| AdvAD+DGI† | **3.97** | **100.0** | 0.34 | 63.60 | **0.9997** | 0.2317 |
| AdvAD-X† | 4.05 | **100.0** | 0.34 | 63.62 | **0.9997** | **0.2301** |

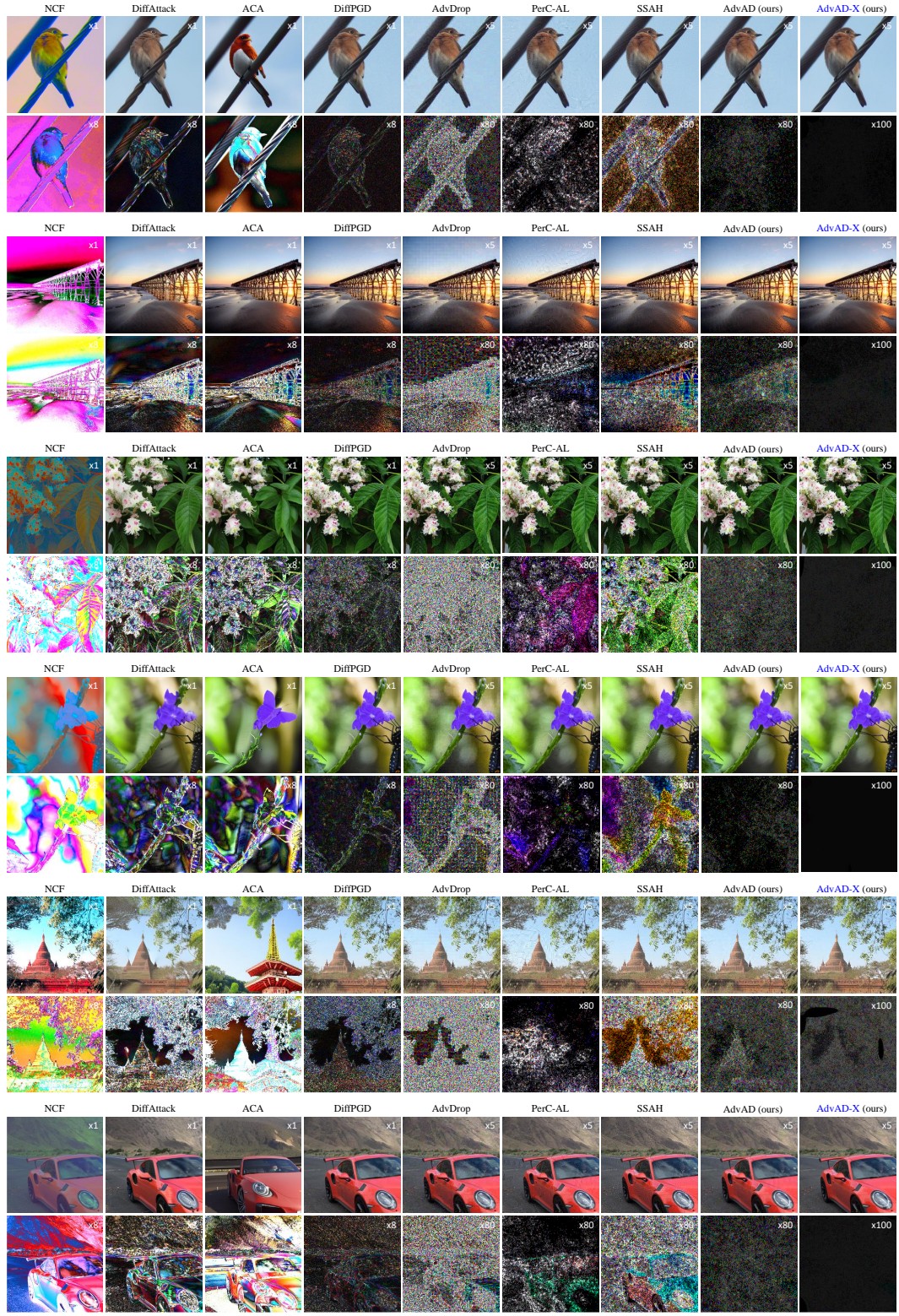

Figure 6: Additional imperceptible adversarial examples and corresponding perturbations generated by various methods.Perturbations are amplified and shown for the convenience of observation. Please zoom in to observe the details of the images.

### D.4 Additional Discussions

**Discussion on Proposition 1.** In the previous sections, we obtained Proposition 1 through extensive derivations, which reformulates the AdvAD attack process using $\lambda_t$ and $\boldsymbol{\delta}_t$. While this formulation does not represent the actual attack procedure, it enables post-analysis after the completion of attacks. In Proposition 1, although the upper bound of $\boldsymbol{\delta}_t$ is theoretically independent of the step $t$, both $\boldsymbol{\delta}_t$ and the coefficient $\lambda_t$ gradually decrease with $t$ in quantitative results of Figure 5 due to the unique properties of AdvAD. Thus, it emerges a hypothesis that whether modifying the coefficient of gradient term of traditional attacks like PGD to decay incrementally could also achieve the imperceptibility. To isolate the impact of this hypothesis, we conduct experiments with a modified version of PGD with step size decay as:

$$\boldsymbol{x}_{t-1} = \Pi_\xi\{\boldsymbol{x}_t + \lambda_t \cdot \eta \cdot \mathrm{sign}(\nabla_{\boldsymbol{x}_t}\mathcal{L}_{\mathrm{CE}}(f(\boldsymbol{x}_t), y_{gt}))\}, \tag{53}$$

where $\lambda_t$ is the same coefficient as in Eq. (39) of Proposition 1 for alignment, and $\eta$ is a fixed small factor for the initial step size. We have searched a lot of values of $\eta$ to determine the optimal range, and the results of attacking three models with different architectures under three typical values of $\eta$ are presented in Table 8.

Table 8: Results of PGD + step size decay strategy and the proposed AdvAD.

| Model | Attack Method | Param. | Time | ASR | $l_\infty$ | $l_2$ | PSNR ↑ | SSIM ↑ |
|---|---|---|---|---|---|---|---|---|
| ResNet-50 | PGD + Step size decay in Eq. (53), $\eta$ = 5e-5 | $T$=1000, $\xi$=8/255 | 2272 | **99.9** | 0.016 | 1.80 | 46.75 | 0.9947 |
| | PGD + Step size decay in Eq. (53), $\eta$ = 3e-5 | | 2228 | 99.0 | **0.008** | 1.17 | 50.41 | 0.9974 |
| | PGD + Step size decay in Eq. (53), $\eta$ = 1e-5 | | 2306 | 7.1 | - | - | - | - |
| | AdvAD (ours) | | 2201 | 99.7 | 0.010 | **1.06** | **51.84** | **0.998** |
| Swin-Base | PGD + Step size decay in Eq. (53), $\eta$ = 5e-5 | $T$=1000, $\xi$=8/255 | 8725 | 98.0 | 0.008 | 1.28 | 49.88 | 0.9975 |
| | PGD + Step size decay in Eq. (53), $\eta$ = 3e-5 | | 8728 | 89.1 | **0.004** | **0.94** | **52.47** | **0.9985** |
| | PGD + Step size decay in Eq. (53), $\eta$ = 1e-5 | | 8715 | 3.9 | - | - | - | - |
| | AdvAD (ours) | | 9729 | **100** | 0.013 | 1.19 | 50.57 | 0.9978 |
| VisionMamba-Small | PGD + Step size decay in Eq. (53), $\eta$ = 5e-5 | $T$=1000, $\xi$=8/255 | 6350 | 89.2 | 0.008 | 1.63 | 47.76 | 0.9959 |
| | PGD + Step size decay in Eq. (53), $\eta$ = 3e-5 | | 6393 | 78.3 | **0.004** | **1.10** | **51.05** | **0.9979** |
| | PGD + Step size decay in Eq. (53), $\eta$ = 1e-5 | | 6348 | 2.5 | - | - | - | - |
| | AdvAD (ours) | | 6154 | **99.7** | 0.016 | 1.62 | 47.94 | 0.9960 |

It can be observed that for PGD with this strategy, the ASR is clearly proportional to $\eta$, the imperceptibility is inversely proportional to $\eta$. However, regardless of how $\eta$ is adjusted, this strategy can not simultaneously match AdvAD in both ASR and imperceptibility. Firstly, for $\eta$ = 5e-5, when attacking VisionMamba, ASR of this strategy is 10.5% lower than AdvAD with close PSNR, and the strategy has a 0.2% higher ASR but a 5.09 dB lower PSNR for ResNet50. For $\eta$ = 3e-5, the ASR against VisionMamba and Swin further degrade, being 10.9% and 21.4% lower than AdvAD, respectively. Finally, for $\eta$ = 1e-5, the modified PGD with step size decay fails to attack all the models. Nevertheless, although this step size decay strategy performs worse than our AdvAD, it indeed enhances the imperceptibility of attacks compared to the original PGD in some cases, which further validates our motivation and modeling approach. This is because, while this strategy follows a completely different technical route than AdvAD, it similarly uses subtler perturbations to progressively push adversarial examples closer to the model's decision boundary. To this end, we leave further research on the potential of this strategy to future work.

**Limitation.** As the primary focus of AdvAD is the imperceptibility, although it achieves better transferability at lower perturbation strength compared with other restricted imperceptible attacks, its transferability is inevitably weaker than other black-box attack methods that operate in larger perturbation spaces and are specifically designed for transferability (like the unrestricted ones). However, the proposed AdvAD is essentially a general attack paradigm with a novel modeling approach and a solid theoretical foundation. By relaxing the constraint of perturbation strength and incorporating enhanced designs for the transferability into the proposed framework of non-parametric diffusion process, AdvAD also has significant potential to be modified into a specific black-box attack, and we also leave this aspect for future research.

