# OpenReview forum: "AdvAD: Exploring Non-Parametric Diffusion for Imperceptible Adversarial Attacks"
_NeurIPS.cc/2024/Conference — NeurIPS 2024 poster_

### Official Review · Reviewer_ytMn · 2024-07-03

**Soundness:** 3
**Presentation:** 3
**Contribution:** 3
**Rating:** 4
**Confidence:** 4

**Summary:**

In this paper, a diffusion-based adversarial attack method is proposed and extensive experiments were conducted to show the effectiveness of the method.

**Strengths:**

1. the paper is well-written and easy to read.
2. both adversarial attack and diffusion model are hot topics.
3. both Linf and L2 results are given in the experiment.
4. An ablation study on T is provided
5. The experiment's results are convincing.

**Weaknesses:**

1. because this paper proposed a new algorithm, at least a pseudo code should be provided to show how the algorithm works.
2. In the table1, it shows that the performance of the model measured in FID sometimes does not match that in other metrics measuring similarity, e.g., psnr, ssim. For example, in the experiment on visionmemba, the FID of the proposed method is not the smallest, as that in other similar metrics. Can the author give some explanation about this?

**Questions:**

same

**Limitations:**

same

---

> ### Author Rebuttal · Authors · 2024-08-07
>
> Thanks for your time in processing our manuscript and the valuable feedback. Our point-by-point responses are as follows.
>     &nbsp;
> **Weakness 1: Pseudo code.**
> **Re:** We appreciate your attention to this detail. We acknowledge that due to page limitations, we did not include the pseudo code in our initial submission. Instead, we made every effort to optimize our presentation and used clear illustrations with corresponding equation numbers (Figure 1) to introduce the workflow of our algorithm. We apologize for any difficulties this may have caused in understanding our manuscript. We have provided the complete pseudo code of our AdvAD and AdvAD-X in the attached PDF of the global author rebuttal, and it has also been included in the revised manuscript to enhance the clarity. Additionally, we have submitted runnable source code in the Supplementary Material for reproducibility, and we will also release the code after this work is accepted for publication.
>
> **Weakness 2: The results of FID are not always the best.**
> **Re:** It can be observed that the FID results of AdvAD are the second lowest only when attacking ConvNeXt and VisionMamba, with SSAH having the lowest results in these cases. Upon careful examination, we consider this to be due to the poor attack performance of SSAH on these two models, for the following reasons: Firstly, please note that the ASRs of SSAH are only **84.6%** and **49.8%** (ours are **100%** and **99.7%**) when attacking these two backbones (as shown in Table 1). As a method for directly optimizing the feature space in imperceptible attacks, SSAH shows such a low attack success rate, indicating that the added adversarial perturbations have minimal impact on the high-dimensional features within the attacked model layers. In other words, when SSAH attacks fail, the high-dimensional features of the crafted adversarial examples and the original images should be closer. Meanwhile, FID measures the Fréchet distance between the high-dimensional features of adversarial examples and original images extracted by a pre-trained Inception-V3. Therefore, the features of the adversarial examples generated by SSAH and the original images in the Inception-V3 model should also be closer, resulting in a lower FID score compared to ours. When all the attacks work normally, due to the inherently lower perturbation strength of AdvAD, the FID score, along with other metrics, achieves the best results among the comparison attacks including SSAH, demonstrating the effectiveness of our novel modeling approach.

---

> > ### Author Response · Authors · 2024-08-12
> > **Experimental verification regarding the previous response to the comment on FID metric (Weakness 2).**
> >
> > To further validate the explanation provided above and offer a better understanding, we directly calculate the cosine similarity of the extracted global features between all adversarial examples and their corresponding original images. The average results for the two backbones are shown in the last two rows of the following table, respectively.
> >
> > | Metric                             | ConvNeXt |            | VisionMamba |            |
> > |------------------------------------|----------|------------|-------------|------------|
> > |                                    | **SSAH**     | **AdvAD** (ours) | **SSAH**        | **AdvAD** (ours) |
> > | **ASR (%) $\\uparrow$**                        | 84.6     | 100.0        | 49.8        | 99.7       |
> > | **$l_2$ $\\downarrow$**                             | 2.24     | 1.49       | 1.95        | 1.62       |
> > | **FID $\\downarrow$**                            | 3.04     | 5.07       | 2.08        | 3.67       |
> > | **Avg. Feature Similarity (Attacked Model)** | 0.5745   | 0.4184     | 0.5424      | 0.2321     |
> > | **Avg. Feature Similarity (Inception)** | 0.9908  | 0.9845     | 0.9938      | 0.9888     |
> >
> > It can be more easily observed that, consistent with our previous explanation, the lower FID results of the SSAH attack are due to its ineffectiveness in attacking the ConvNeXt and VisionMamba backbones. As shown in the sixth row of the table, despite the higher strength of the perturbation injected into the adversarial examples (as indicated by the $l_2$​ metric), the feature similarity of SSAH remains higher than ours. This suggests that the adversarial perturbation crafted by SSAH is more difficult to impact the high-dimensional features on which the model's final classification decision relies, leading to its lower ASR. For the Inception model, which is used to calculate the FID metric, SSAH's feature similarity is anomalously further amplified to above 0.99 (the last row), indicating that the adversarial perturbation crafted by SSAH has almost no effect on the high-dimensional features of the clean images for the Inception model, resulting in its lower FID results.
> > However, when all attacks are performing correctly, the magnitude of changes in image-space and feature-space can be considered as positively correlated in the scenario of imperceptible attacks. In this case, benefiting from the inherently lower perturbation strength of our modeling approach, AdvAD consistently achieves the best FID score when attacking the other models, as shown in Table 1 and Table 5 (Appendix).

---

> ### Author Response · Authors · 2024-08-13
> **Respectful Request for Further Discussion**
>
> Dear Reviewer ytMn,
>
> We sincerely thank you for your valuable feedback and the time you have taken to process our submission. As the discussion phase is nearing its end, we respectfully ask for your help once again to review our responses and let us know if they address your concerns.
>
> Following your valuable suggestions, we have included the pseudo code for our AdvAD and AdvAD-X in the attached PDF of the global author rebuttal, and it has also been incorporated into the revised version to further enhance clarity. Additionally, we have provided a detailed explanation and corresponding experimental verification for the FID results, which you expressed concerns about.
>
> Please also let us know if you have any further questions about this paper. We have made every effort to enhance our work based on your insightful comments, and we would deeply appreciate it if you could further support us!
>
> Best regards,
> The authors

---

### Official Review · Reviewer_nUXD · 2024-07-03

**Soundness:** 3
**Presentation:** 3
**Contribution:** 2
**Rating:** 5
**Confidence:** 3

**Summary:**

This work proposes a novel adversarial attack framework called Adversarial Attacks in Diffusion (AdvAD). Unlike prior methods that rely on generative models or specific loss functions, AdvAD formulates attacking as a non-parametric diffusion process. This approach theoretically explores a fundamental modeling strategy instead of leveraging the denoising or generation capabilities of diffusion models with neural networks. AdvAD iteratively refines the attack by crafting subtle adversarial guidance based solely on the targeted DNN, without requiring any additional network. This process progressively transforms the original image into an imperceptible adversarial example.

**Strengths:**

1.	The concept of leveraging the core principles of diffusion models to craft adversarial examples is intriguing.
2.	The writing style in the Abstract and Introduction sections is engaging.
3.	The effectiveness of AdvAD is validated through experiments.

**Weaknesses:**

While I have carefully reviewed the work several times, some uncertainties remain. These are addressed in the following "Questions" section. Due to these uncertainties, the initial recommendation is a borderline decision, which can be revised upwards or downwards based on the authors' response.

**Questions:**

1.	The paper claims that AdvAD achieves superior imperceptibility compared to other methods. However, after multiple readings, the justification for this remains unclear. While the text mentions "much subtler yet effective adversarial guidance at each step", "grounded on diffusion model's theoretical foundation", and various diffusion-related formulations, it lacks a clear explanation (or theoretical verification) of how using non-parametric diffusion leads to attacks with lower perturbations.
2.	Based on my understanding, AdvAD shares similarities with traditional attacks like PGD, but with a fixed number of optimization iterations (controlled by the total diffusion model timesteps T) and a gradually decaying learning rate (controlled by the $α_t$ value). I hypothesize that the decayed learning rate is a key factor contributing to imperceptibility. The learning rate starts low (at the beginning of T steps) and gradually decreases to 0 (at the end of 0 steps). This suggests that small perturbations are added first, followed by even smaller ones, potentially leading to adversarial examples closer to the classification boundary and thus exhibiting higher imperceptibility. Could the authors comment on the validity of this hypothesis?
3.	Following the previous question, to isolate the impact of the decayed learning rate strategy, it would be beneficial to include comparative experiments. These experiments would directly apply a small learning rate initialization and a decayed learning rate schedule to a traditional attack algorithm. If this approach achieves similar performance to AdvAD, it would suggest that the emphasis on diffusion model theory might not be necessary. If not, the authors should highlight the specific benefits of the diffusion model mechanism that contribute to imperceptibility.
4.	Does the selection of the initial Gaussian noise sample $ε_0$ affect the performance of AdvAD?
5.	In Table 1, is the FID metric computed between the perturbed images and the "imagenet-compatible dataset"? If the goal is to assess the realism of the adversarial examples compared to natural images, a more comprehensive evaluation would be to compare them with the raw ImageNet images. The "imagenet-compatible" dataset, with only 1000 images, might not adequately capture the entire distribution of natural images.
6.	Line 248 states that "the optimizer usually cannot find the global optimal solution, and optimization-based methods tend to show sub-optimal ASR." How can AdvAD achieve better ASR compared to these optimization-based methods?
7.	In Table 2, which classifier was used to generate the adversarial samples? The robustness of the adversarially trained model Inc-V3_{adv} and the ensemble model Inc-V3_{ens4} appears very low (almost 100% ASR). Additionally, were Res-50, Swin-B, and ConvNeXt-B clean classifiers or adversarially trained?
8.	Line 292 suggests that "a larger T exhibits better imperceptibility, while a smaller T implies stronger black-box transferability." Can the authors elaborate on this trade-off?
9.	There are a few minor writing errors throughout the paper, such as "a' adversarial" (Line 140), "fistly" (Line 144), and "an' scheme" (Line 191).
10.	There are specific questions regarding the proofs of theorems and propositions in the Appendix:

(1)	In the second row equality of Eq. 6, shouldn’t $\epsilon_T$ be $\epsilon_{T+1}$ in the first term on the numerator of the first term?

(2)	In Line 546, it Is clear that $\epsilon_{T+1}$ satisfies the inequality as $\epsilon_{T+1}$ equals $\epsilon_0$, but how does this hold for $\epsilon_{T}$ as well?

(3)	Eq. 20, third row inequality: How is this inequality derived?

**Limitations:**

Yes, limitations are identified in the paper.

---

> ### Author Rebuttal · Authors · 2024-08-07
>
> Thanks for your time in processing our manuscript and the valuable feedback. Our point-by-point responses are as follows.
>
> **Question 1:  Explanation and verification of AdvAD.**
> **Re:**
> 1) **Intuitive explanation.** The superior imperceptibility of AdvAD first comes from its modeling philosophy inherited from diffusion models. For image generation tasks, one of the widely recognized advantages of diffusion models lies in breaking down the complex problem of directly generating images (e.g., GANs) into a series of simpler tasks, gradually pushing a Gaussian distribution toward the image distribution by approximating the noise at each diffusion step, thus to significantly reduce the difficulty of generation with a divide-and-conquer manner. Similarly, compared to previous attack frameworks that directly optimize or inject perturbations into images, the proposed _non-parametric diffusion process_ allows AdvAD to decompose attacking into the relatively simpler task of imposing _adversarial guidance_ by manipulating diffusion noise at each diffusion step to progressively guide the endpoint of diffusion from the original image to a desired adversarial example. Thus, the adversarial guidance injected at each step could be subtler but more effective, ultimately achieving imperceptible attacks with inherently lower perturbation strength.
>
> 2) **Theoretical verification.** As a completely novel attack paradigm without any loss function, optimizer, or additional neural network, the essence of AdvAD exactly lies in the _subtler_ (for imperceptibility) yet _more effective_ (for attack efficacy) adversarial guidance, and both the characteristics are theoretically grounded.  Firstly, the conditional sampling technique enables diffusion models to sample from a given conditional distribution (Eq. 3). Derived from it, the proposed AMG module treats the goal of attack as a condition and gradually moves the original diffusion trajectory of the image towards this adversarially conditional distribution through a non-parametric diffusion process (Eq. 4-8), while the PC module synergistically ensures the process is streamlined and controllable (Theorem 1), providing guarantee of successful attack. Secondly, beyond the aforementioned modeling philosophy, the imperceptibility is also supported in the unique property of adversarial guidance as described in line 156-161. In addition, our Proposition 1, 2 and experimental results (especially for Sec. 4.5) provide further analysis and validation of this point.
>
> **Question 2, 3: Hypothesis and PGD with decaying learning rate.**
> **Re:**
> 1) As shown in Eq. 9, the $l _\\infty$ upper bound of the guidance strength is irrelative with step $t$. This means that there is no 'decaying learning rate/step size' coefficient for the guidance during the non-parametric diffusion process, and no loss function (e.g., CrossEntropy) is used as in traditional algorithms for gradient ascent. We assume your hypothesis originates from the two decreasing curves about Proposition 1 in Figure 4. Actually, Proposition 1 is an analytical conclusion drawn from extensive formula derivations in Appendix B.3, which demonstrates that the effect of adversarial guidance on the entire process diminishes from strong to weak. This is because, due to the nature of the non-parametric diffusion process, the earlier the guidance is injected, the greater its cumulative impact on the subsequent steps, corresponding to more movement distance toward the target distribution for the noisy sample at that step.
>
> 2) We conduct experiments of the PGD+decaying step size strategy. We modify the PGD attack as: $x ^{t+1} = \\Pi(x ^t + \\lambda _t \\cdot \\eta \\cdot sign(\\nabla _{x _t}L _{CE}(x ^t, y)))$, where $T=1000$, $\\lambda _t$ is consistent with our Proposition 1 for alignment,  and $\\eta$ is a fixed small factor for the initial learning rate. We explored a lot of $\\eta$ values to determine the optimal range, and the results attacking three models with different architectures with three typical $\\eta$ are presented in Table R2 of the attached PDF. It can be observed that for PGD with this strategy, the ASR is clearly proportional to $\\eta$, while the imperceptibility is inversely proportional to $\\eta$. However, regardless of how $\\eta$ is adjusted, this strategy _can not_ simultaneously match AdvAD in both ASR and imperceptibility. Firstly, for $\\eta$ = $5\\times10^{-5}$, when attacking VisionMamba, ASR of this strategy is 10.5% lower than AdvAD with similar PSNR. For ResNet50, the strategy has a 0.2% higher ASR but a 5.09 dB lower PSNR. When $\\eta$ = $3\\times10^{-5}$, the ASR against VisionMamba and Swin further degrade, being 10.9% and 21.4% lower than AdvAD, respectively. When $\\eta=1\\times10^{-5}$, this method fails to attack all the models.
>
> **Question 4: Initial Gaussian noise.**
> **Re:** All Gaussian noises are randomly initialized and have almost no impact because they merely serve as a medium of the injected adversarial guidance. With various seeds, the effect on the final results is akin to slight random fluctuations, at ±0.1% for ASR and ±0.02 for PSNR.
>
> **Question 5: Calculation of FID.**
> **Re:** For adversarial attacks, our goal is to assess the similarity between adversarial examples and their corresponding clean images to measure the imperceptibility of attacks, rather than the gap with the natural image distribution. Therefore, we adopt the FID calculation method consistent with previous work. Moreover, in Table 1, we have included results of the MUSIQ metric, which assesses the realism of adversarial examples without reference images.
>
> **Due to the character limit, point-by-point responses to the remaining questions are placed in Part 2 of the global author rebuttal. Please kindly refer to that section for details.**

---

> > ### Comment · Reviewer_nUXD · 2024-08-11
> > **Thanks for the response.**
> >
> > After reviewing the authors' response and considering the innovative nature of the attack they have implemented, I have decided to raise my score to 5.

---

> > > ### Author Response · Authors · 2024-08-11
> > >
> > > Thank you for the response. We sincerely appreciate your valuable comments and are grateful for the recognition of our work.

---

### Official Review · Reviewer_jCt2 · 2024-07-10

**Soundness:** 3
**Presentation:** 3
**Contribution:** 2
**Rating:** 4
**Confidence:** 4

**Summary:**

This paper proposes a new method, AdvAD, for generating imperceptible adversarial attacks against deep neural networks (DNNs). AdvAD is based on a novel non-parametric diffusion process, which initializes a fixed diffusion noise and then manipulates it at each step using adversarial guidance crafted by two modules: Attacked Model Guidance (AMG) and Pixel-level Constraint (PC). This process gradually leads the image from its original distribution to a desired adversarial distribution. The paper also introduces an enhanced version, AdvAD-X, which aims to achieve extreme performance under an ideal scenario. Extensive experiments demonstrate the effectiveness of both AdvAD and AdvAD-X in terms of attack success rate, imperceptibility, and robustness compared to state-of-the-art methods.

**Strengths:**

The paper's main strength lies in its novel approach to generating imperceptible adversarial attacks. The use of a non-parametric diffusion process is innovative and shows promising results in terms of attack effectiveness and imperceptibility. The paper is well-written and easy to follow, with clear explanations of the proposed method and experimental setup.

**Weaknesses:**

The paper has several weaknesses. The technical novelty is limited, as the core idea of using diffusion models for adversarial attacks has been explored in prior work (e.g., DiffAttack, ACA). The paper lacks a clear comparison with these existing diffusion-based attack methods, making it difficult to assess the incremental contribution of AdvAD.

Another thing is how much cost is it really to generate adversarial examples this way. I mean, one important use of generating adversarial examples is for adversarial training. So if it becomes really expensive to generate adversarial examples, such a use wouldn't be really meaningful. Then what is the use of these adversarial examples really?

**Questions:**

What are the specific differences between AdvAD and existing diffusion-based attack methods like DiffAttack and ACA? A detailed comparison would help clarify the novelty of the proposed approach.

What are the potential defensive measures against AdvAD, and how can they mitigate the threat posed by this attack?

How does the computational complexity of AdvAD compare to other state-of-the-art methods, especially for large-scale datasets or models?

---

> ### Author Rebuttal · Authors · 2024-08-07
>
> Thanks for your time in processing our manuscript and the valuable feedback. Our point-by-point responses are as follows.
>     &nbsp;
> **Weakness 1 & Question 1: Clear and detailed comparison with existing diffusion-based attack methods.**
> **Re:** Compared to other recent notable works exploring adversarial attacks based on diffusion models, the proposed AdvAD is a completely novel approach distinct from existing attack paradigms. It is the first pilot framework which innovatively conceptualizes attacking as a non-parametric diffusion process by theoretically exploring fundamental modeling approach of diffusion models rather than using their denoising or generative abilities, achieving high attack efficacy and imperceptibility with intrinsically lower perturbation strength. To further highlight and clarify our contributions, we have provided a detailed comparison below between AdvAD and other diffusion-based attack methods in various aspects.
> 1) **Motivation.** Existing diffusion-based attacks achieve imperceptibility by utilizing the denoising or generative abilities of diffusion models to heavily modify texture or semantic attributes of images in an image edition-like way rather than inject typical adversarial noises. In contrast, the proposed AdvAD aims to address the fundamental challenge of imperceptible attacks, that is, performing attack with inherently minimal perturbation strength from a modeling perspective.
>
> 2) **Theory of Attack.** Although diffusion models are introduced in the existing diffusion-based attacks, their theoretical foundations still adhere to prior attack paradigms, either by integrating the optimization of adversarial loss functions (e.g., cross-entropy) or classical attack methods (e.g., PGD) with nested loops of forward and backward diffusion steps. In comparison, we propose a novel non-parametric diffusion process for performing attacks as a new paradigm. The modeling of AdvAD is theoretically derived from conditional sampling of diffusion models, supporting its attack performance and imperceptibility, and does not require any loss functions, optimizers, or additional neural networks.
>
> 3) **Methodology.** Among the existing diffusion-based attacks, DiffAttack and ACA combine the optimization of adversarial losses with the Stable Diffusion to generate unrestricted adversarial examples, while Diff-PGD and AdvDiffuser incorporate the classic PGD attack into the normal diffusion process. Furthermore, as typical diffusion models, these attacks require using pre-trained neural networks to estimate the noise term at each diffusion step. For our AdvAD, the adversarial example is crafted via an uni-directional non-parametric diffusion process from an initialized and fixed diffusion noise to final sample. At each step, a much subtler (for imperceptibility) yet more effective (for attack performance) adversarial guidance is calculated via the proposed AMG and PC modules without additional networks, then injected by manipulating the diffusion noise.
>
> 4) **Effectiveness.** Due to the uncertainty of generative models and the unrestricted attack setting, existing diffusion-based attacks inevitably cause unnatural and unreasonable artifacts in the crafted adversarial examples, especially for images with complex content. Benefiting from our modeling approach, the proposed AdvAD is able to apply precise control over the perturbations and accomplish imperceptible attacks with inherently low perturbation strength. Extensive experimental results have also demonstrated the superiority of AdvAD.
>
> Based on AdvAD, we further propose an enhanced version AdvAD-X to explore the extreme performance in an ideal scenario for the first time, which also possesses theoretical significance and provides new insights for revealing the robustness of DNNs.
>
> **Weakness 2 & Question 3: Computational complexity.**
> **Re:** We respect and concur with your point of view. One of the important roles of adversarial attacks is to promote corresponding defense methods and enhance the robustness of DNNs, and algorithm efficiency is a key factor to consider in practice of adversarial training.
>
> In column 3 of Table 1, we have reported the total time cost of each attack for processing 1000 natural images. Thanks to our novel non-parametric and uni-directional diffusion process, which does not require additional networks or nested loops of forward and backward diffusion steps, the computational complexity of AdvAD is mainly concentrated in taking partial derivatives to obtain adversarial guidance (note that this is not the gradient of loss function) at each step. Thus the cost of AdvAD is significantly lower than ACA and DiffAttack, which contain Stable Diffusion within loops of optimization. For Diff-PGD, it is more time-consuming than AdvAD when attacking small models (ResNet-50, 1.5x) but slightly faster for larger models (Swin-Base, 0.8x). Additionally, for non-diffusion optimization-based attacks, the computational complexity of AdvAD is lower than PerC-AL and comparable to the others, while the performance of AdvAD is much better.
>
> **Question 2: Potential defensive measures against AdvAD.**
> **Re:** In our research, we have considered the robustness of AdvAD against defense methods and presented the experimental results in Section 4.3. It can be observed that due to the inherently low perturbation strength of AdvAD, it is relatively more susceptible to defense methods based on purification, which is similar to other state-of-the-art restricted imperceptible attack methods. However, the results indicate that AdvAD can relatively easily bypass robust models. We hypothesize that this is because these robust models typically use previous attack algorithms based on adversarial losses or gradient ascent for adversarial training, whereas AdvAD, as a new attack paradigm, is not included. Thus, we also anticipate that AdvAD will further advance the development of related defense fields.

---

> ### Author Response · Authors · 2024-08-13
> **Respectful Request for Further Discussion**
>
> Dear Reviewer jCt2,
>
> We sincerely thank you for your valuable feedback and the time you have taken to process our submission. As the discussion phase is nearing its end, we respectfully ask for your help once again to review our responses and let us know if they address your concerns.
>
> In brief, we have provided a detailed point-by-point comparison with other diffusion-based methods from various perspectives to further complement and highlight the significant innovations and contributions of our paper, in addition to those you kindly recognized in the Strengths section. Moreover, we have elaborated on the computational complexity and potential defense methods following your constructive suggestions. It is our pleasure to receive your feedback.
>
> Please also let us know if you have any further questions about this paper. We have made every effort to enhance our work based on your insightful comments, and we would deeply appreciate it if you could further support us!
>
> Best regards,
> The authors

---

### Official Review · Reviewer_ygiL · 2024-07-12

**Soundness:** 3
**Presentation:** 3
**Contribution:** 3
**Rating:** 6
**Confidence:** 4

**Summary:**

In this paper, the authors propose the Adversarial Attack in Diffusion method called AdvAD, which crafts imperceptible perturbations from the model perspective without the need for additional networks.

During the non-parametric diffusion process, the proposed AdvAD method introduces Attacked Model Guidance (AMG) and Pixel-level Constraint (PC) modules to ensure both imperceptibility and transferability of the adversarial examples.

Moreover, an enhanced version AdvAD-X is further introduced to achieve better imperceptibility and lower computational complexty.

Experiments conducted on CNN, ViT, and Mamba architectures using the ImageNet benchmark dataset demonstrate that the proposed AdvAD method achieves superior performance compared to baseline methods in terms of attack success rate, imperceptibility, and robustness.

**Strengths:**

1. The proposed AdvAD method reformulates the generation of imperceptible adversarial examples as a non-parametric diffusion process, effectively eliminating the negative effects caused by the uncertainty of generative models.

2. The proposed AMG and PC modules inject adversarial guidance progressively at each step without the need for additional networks, and they have a solid theoretical guarantee that the error caused by approximation is strictly bounded.

3. Experimental results verify the effectiveness of the proposed AdvAD method, which achieves higher attack success rate, lower perturbation strength, and better image quality.

**Weaknesses:**

1. In section 3.4, the details of AdvAD-X are not explained clearly. The exact step of adopting original diffusion noise in the DGI strategy is not discussed. Additionally, in the CA strategy, the definition of non-critical image regions is not introduced.

2. In section 4.4, the transferability experiment lacks comprehensiveness. As shown in Table 3, the proposed AdvAD method is only compared to a limited number of baseline methods.

3. In Table 3, with a limited budget of updating iterations, the proposed AdvAD (T=10) performs very similarly to the classic PGD method when using Mob-V2 as a surrogate model, improving the black-box ASR by less than 1%.

4. The experimental settings for Table 4 are not fully introduced, such as the surrogate model used for crafting adversarial examples.

**Questions:**

1. How are non-critical image regions defined for AdvAD-X? Is this definition universally applicable to all images, or is it adjusted for each image separately?

2. Why were other attack methods (such as NCF, ACA, Diff-Attack, Diff-PGD, etc.) not considered in the transferability experiment shown in Table 3?

3. As only a few values (e.g., 10, 100, 1000) are discussed, how can the optimal value of the step $T$ be chosen to ensure both imperceptibility and transferability in practice?

**Limitations:**

The proposed AdvAD-X method is only suitable for ideal scenarios where the crafted adversarial example is directly input to DNNs without undergoing quantization.

---

> ### Author Rebuttal · Authors · 2024-08-07
>
> Thanks for your time in processing our manuscript and the valuable feedback. Our point-by-point responses are as follows.
>     &nbsp;
> **Weakness 1 & Question 1: Details of AdvAD-X: exact step with DGI strategy, definition of non-critical image regions.**
> **Re:** Due to the page limitation of the main paper, we placed the ablation study results of AdvAD-X in **Appendix C.3**, including the number of iterations of AdvAD-X (column 2, Table 6). For your convenience, we transcribes it here as follows:
>
> | Attack | Iter. | ASR ($\\uparrow$) | $l_2$ ($\\downarrow$) | PSNR ($\\uparrow$) | SSIM ($\\uparrow$) | FID ($\\downarrow$) |
> | :-------- | :-------: | :-------: | :------: | :-------: | :-------: | :------: |
> | AdvAD$\\textcolor{blue}{\\dagger}$            | 1000 | **100.0** | 0.97 | 52.60 | 0.9984 | 2.3894 |
> | AdvAD + CA$\\textcolor{blue}{\\dagger}$   | 1000 | **100.0** | 0.89 | 53.27 | 0.9987 | 2.2033 |
> | AdvAD + DGI$\\textcolor{blue}{\\dagger}$ | **3.97**  | **100.0** | **0.34** | 63.60 | **0.9997** | 0.2317 |
> | AdvAD-X$\\textcolor{blue}{\\dagger}$        | 4.05  | **100.0** | **0.34** | **63.62** | **0.9997** | **0.2301** |
>
> From AdvAD to AdvAD-X,  the effect of the two strategies of CA and DGI are shown in this table. It can be observed that adding CA in each step of AdvAD slightly improves impercepbility while maintaining the attack success rate of 100\%. However, the DGI strategy significantly reduces the iterations of performing AMG and PC from 1000 to 3.97, which indicates that our framework theoretically only requires very little injected  adversarial guidance to successfully perform attacks, proving the performance of our modeling method as well as the effectiveness of the adversarial guidance. In AdvAD-X, which finally uses both CA and DGI, the guidance strength in each step is further suppressed, resulting in a slight increase in the total number of iterations required adaptively, but the final perturbation strength continues to decrease to a more extreme level.
>
> For the CA strategy, we adopt the GradCAM (Gradient-weighted Class Activation Mapping) to obtain mask about non-critical image regions for each image separately (as mentioned in lines 201-204 of the main paper). GradCAM is a widely used technique to indicate important regions of the input image that a classification model focuses on when making decisions about a particular category. Specifically, we  calculate the heatmap via GradCAM as a mask $\\boldsymbol{m}$ with the same resolution of image. In this mask , each pixel value ranges from 0 to 1 with lower score representing lower importance, thus the non-critical regions are defined. Consequently, AdvAD-X suppresses the intensity of injected adversarial guidance at each step by performing element-wise multiplication with the mask $\\boldsymbol{m}$, which can be expressed as:
> $$\\boldsymbol{\\hat{\\epsilon}}_t = \\boldsymbol{\\epsilon} _0 - \\boldsymbol{m} \\cdot \\sqrt{1-\\alpha_t} \\nabla _{\\boldsymbol{\\hat{x}}_t}\\text{log}(1-p _f(y _{gt}|\\boldsymbol{\\hat{x}} _{t}^{0}))$$
> Additionally, we have provided the pseudo code for the AdvAD and AdvAD-X algorithms in the PDF file of the global author rebuttal for better understanding. We also hope the discovery that AdvAD-X can successfully attack with extremely low perturbation strength with floating-point raw data can bring new inspiration to reveal the robustness and interpretability (e.g., decision boundaries) of DNNs.
>
> **Weakness 2 & Question 2: Other attack methods (NCF, ACA, Diff-Attack, Diff-PGD) are not considered in Table 3.**
> **Re:**
> 1) Rather than only focusing on transferability, the purpose of Table 3 is to show that although one of the main advantages of the proposed method is the imperceptibility achieved by the novel modeling approach with inherently low perturbation strength, our AdvAD also surpasses other state-of-the-art restricted imperceptible attack methods in transferability, even at much smaller perturbation cost. This is despite the evident positive correlation between transferability and perturbation strength as described in line 292-295. Additionally, more than a comparison of transferability, Table 3 also serves as an ablation study to illustrate the effect of step $T$, demonstrating AdvAD could be flexibly adjusted through $T$ as a novel and general attack framework.
> 2) The NCF, ACA, and other methods you mentioned are unrestricted attacks, which dose not impose objective limitation on perturbation strength, resulting in significant alterations to the images (as shown in Table 1, their $l _2$ distances are x10-x75 greater than AdvAD) and inevitably producing unnatural artifacts or unreasonable semantic modifications sometimes (as visualizations in Figure 2 and 5). Given this substantial difference in perturbation cost, comparing the transferability of unrestricted and restricted methods is inappropriate. Therefore, we did not include these methods in Table 3. Instead, as aforementioned, we compared our method with other state-of-the-art restricted imperceptible attacks and demonstrated that our approach surpasses them in both imperceptibility and transferability, simultaneously.
>
> **Weakness 3 & Question 3: Similar performance to PGD using Mob-V2 as a surrogate model, and the optimal value of $T$.**
> **Re:**  In the global author rebuttal, we have further discussed transferability of the proposed AdvAD and provide more experimental results to demonstrate its effectiveness, including more values of $T$, the optimal value selection, more comprehensive comparisons with other attacks, etc. Please kindly refer to the corresponding section for more details.
>
> **Weakness 4: Experimental settings of Table 4.**
> **Re:**   We apologize for this omission. The attacked model adopted in Table 4 is the classic ResNet50, and the total step $T$ is consistently set to 1000 for all $\\xi$. We have added the above description of the experimental setting in the revised manuscript.

---

> > ### Comment · Reviewer_ygiL · 2024-08-12
> >
> > Thanks for the authors’ responses. My concerns have been properly addressed.

---

> > > ### Author Response · Authors · 2024-08-12
> > >
> > > Thank you for your feedback, and we are glad that we have addressed your concerns.

---

### Official Review · Reviewer_K1mG · 2024-07-22

**Soundness:** 3
**Presentation:** 3
**Contribution:** 3
**Rating:** 6
**Confidence:** 3

**Summary:**

This paper studies a task about generating the imperceptible adversarial noise using the diffusion model. The proposed method theoretically models the attack process as a non-parametric diffusion process. Extensive experiments demonstrate the effectives of the proposed method.

**Strengths:**

(1) The writing is good and easy to follow.
(2) The technique novelty is good, which studies the imperceptible adversarial perturbation generation as a non-parametric diffusion process. The behind theory is solid.
(3) Authors conduct extensive experiments and compare many baselines to show the effectiveness of the proposed method.

**Weaknesses:**

(1 ) The adopted defense methods are not strong, lacking some strong baselines, like the random smoothing.
[1] Cohen, Jeremy, Elan Rosenfeld, and Zico Kolter. "Certified adversarial robustness via randomized smoothing." international conference on machine learning. PMLR, 2019.

(2) Could the generated adversarial examples used for black-box (or transfer-based) attack settings? What's the limitation?

**Questions:**

see weakness

**Limitations:**

see weakness

---

> ### Author Rebuttal · Authors · 2024-08-07
>
> Thanks for your time in processing our manuscript and the valuable feedback. Our point-by-point responses are as follows.
>     &nbsp;
> **Weakness 1: Defense of random smoothing.**
> **Re:** In Table 3 of the paper, we have evaluated the robustness of proposed AdvAD using numerous advanced defense methods, including three post-processing purification methods and six adversarial training robust models to defense attacks. Random smoothing is indeed another strong defense method, and we have conducted experiments to test the performance of our attack on it following your suggestion.
> Since random smoothing is not a truly end-to-end model but a method that uses the base model to make multiple predictions on noise-augmented images, we adpot a semi-white-box setup to fully test the attack performance. Specifically, we craft adversarial examples using only the base model without smoothing process, then apply an additional 100 rounds (n=100) of random smoothing voting on the generated adversarial examples and tested the final attack success rate. The results are as follows:
> |         |  &emsp;&emsp; $\\sigma = 0.25$   |    |  &emsp;&emsp; $\\sigma = 0.50$    |    | &emsp;&emsp; $\\sigma = 1.00$   |    |
> |---------|-----------------------------|----|-----------------------------|----|-----------------------------|----|
> |         | **Top-1 Error (%)**    |  $l _2$  &emsp;&emsp;       | **Top-1 Error (%)**             | $l _2$   &emsp;&emsp;        | **Top-1 Error (%)**             | $l _2$  &emsp;&emsp;         |
> | **clean**   | 17.3                      |  /  | 30.3                      |  /  | 46.8                      |  /  |
> | **AdvDrop** | 25.2 (+7.9)             | 5.97 | 33.5 (+3.2)             | 6.21 | 48.7 (+1.9)             | 5.61 |
> | **SSAH**    | 21.8 (+4.5)             | 13.84 | 32.4 (+2.1)            | 14.82 | 46.9 (+0.1)            | 13.68 |
> | **AdvAD**   | **28.2 (+10.9)**             | **2.41** | **36.8 (+6.5)**           | **2.51** | **50.4 (+3.6)**            | **2.08** |
>
> It can be seen that for all random smoothing defenses using base models pre-trained with different variances ($\\sigma$), the proposed AdvAD achieves a higher attack success rate with smaller $l_2$-norm perturbations compared to other state-of-the-art imperceptible attacks (attack ), which further demonstrates the effectiveness of the proposed AdvAD. In the revised version, we will include the relevant references and results.
>
> **Weakness 2: Transferability and limitation.**
> **Re:** Sure. The proposed AdvAD achieves the best imperceptibility with inherently minimal perturbation strength through a novel non-parametric modeling approach, while also surpassing other state-of-the-art restricted imperceptible attacks in transferability at the same time. The transferability of AdvAD is directly related to the step $T$, that is, the imperceptibility is positively correlated with the size of $T$, while transferability is negatively correlated with it. In Table 3 of the original manuscript, we have reported the relevant experimental results, and we have further discussed transferability in more detail in the global author rebuttal section. In Table R1, we provided results corresponding to additional values of $T$, and in Figure R1, we plotted the relationship curve between transferability and imperceptibility to find the optimal trade-off, conducting a more comprehensive evaluation. All experimental results demonstrate the superiority of AdvAD. Please refer to that section for more details.
>
> Regarding limitations, although AdvAD achieves better transferability at lower perturbation strength, its primary focus remains on the imperceptibility of the attack. Consequently, its transferability is inevitably weaker than other  black-box attack methods that operate in larger perturbation spaces and are specifically optimized for transferability. However, the proposed AdvAD is essentially a general attack paradigm with a novel modeling approach and a solid theoretical foundation. Therefore, by relaxing the constraint of perturbation strength and incorporating enhanced designs for the transferability, AdvAD also has significant potential in the direction of black-box attacks. We leave this aspect for future work.

---

> > ### Comment · Reviewer_K1mG · 2024-08-13
> >
> > Thanks for your response. It addresses most of my concerns. Hope authors could include these discussions into the revision.

---

> > > ### Author Response · Authors · 2024-08-13
> > >
> > > Thank you for the response. We appreciate your valuable review and we will incorporate the discussions to the revised version accordingly.

---

### Author Rebuttal · Authors · 2024-08-07

We thank all the reviewers for their valuable feedback and recognition of our contributions. We also appreciate they find the proposed method is novel (**K1mG**, **ygiL**, **nUXD**, **ytMn**), theory is solid (**K1mG**, **ygiL**), paper is well-written (**K1mG**, **jCt2**, **nUXD**, **ytMn**), and the experiments have demonstrated the effectiveness of our AdvAD and AdvAD-X (**K1mG**, **ygiL**, **jCt2**, **nUXD**, **ytMn**).
    &nbsp;
### **Part 1: Responses to the common concerns of step $T$ and transferability of AdvAD.**
We find that three reviewers (**K1mG**, **ygiL**, **nUXD**) are curious about the effect of step $T$ and the transferability exhibited by the proposed AdvAD. Although the primary focus of our method is to achieve imperceptible attacks with inherently minimal perturbation strength from modeling perspective, AdvAD is actually a complete novel and general attack framework that distinct from previous gradient-ascent or optimization-based ones since there are no loss functions, optimizers, or additional neural networks required. By conceptualizing attacking as a non-parametric diffusion process with adversarial guidance, AdvAD can flexibly adjust the decomposition granularity of the diffusion trajectory through the step $T$. A larger $T$ corresponds to a finer decomposition granularity, resulting in subtler yet effective adversarial guidance, which lead to final adversarial examples with lower perturbation strength. Conversely, a smaller $T$ corresponds to a coarser granularity, resulting in higher perturbation strength, which favors transferability. Despite of the contradiction between perturbation strength and transferability, compared to other state-of-the-art restricted imperceptible attacks, AdvAD still demonstrates both better imperceptibility and transferability simultaneously as shown in Table 3 of the manuscript. To further demonstrate the effect, we present five more values of $T$ in Table R1 of the attached PDF.

Additionally, to elaborate the relationship between transferability and imperceptibility of AdvAD, as well as the optimal trade-off in practice, we have plotted two line graph according to Table R1. As shown in Figure R1 (a), as the value of $T$ on the horizontal axis changes, the relationship between imperceptibility and transferability shows a clear proportional trend mentioned above, consistent across different surrogate models. For the optimal trade-off, we consider the intersection point of the two curves to represent a balance between imperceptibility and transferability. Accordingly, for the ResNet-50 and MobileNetV2 models, the optimal values of  $T$ are 50 and 25, respectively. Moreover, Figure R1 (b) illustrates more direct curves of this relationship and the positions of other comparison methods within it. Note that, all the other comparison methods are located to the lower left of the curve of AdvAD. This indicates that our method consistently achieves the best results in both transferability and imperceptibility, fully demonstrating the effectiveness of the proposed AdvAD and the novel non-parametric diffusion based modeling approach. All the supplementary results have been included in the revised manuscript.
    &nbsp;
### **Part 2: Responses to the remaining questions of Reviewer  **nUXD**.**
**Question 6: Why AdvAD achieves better ASR compared to the optimization-based methods?**
**Re:** Optimization-based methods require balancing attack performance and imperceptibility through loss functions and may fall into local optima. In contrast, our method uses inherently small yet effective adversarial guidance to gradually push the non-parametric diffusion towards an adversarially conditioned distribution, without including any penalty terms for imperceptibility, thereby achieving a higher ASR.

**Question 7: Settings of Table 2.**
**Re:** As stated in the first row, the left half of Table 2 employs a standard ResNet50 to generate adversarial examples for three post-processing defense methods, while the right side presents the results of white-box attacks on adversarially trained robust models. Among these, Inc-V3 represents an earlier (2018) classic robust model, whereas the others are state-of-the-art adversarial training models recently published (e.g., [48] at NIPS 2023), thus exhibiting enhanced robustness.

**Question 8, 9: Step $T$ and typos.**
**Re:** We have elaborated on the trade-off issue in Part 1 of this author rebuttal. For writing errors, we thank you for kindly pointing them out, and we have carefully proofread and corrected all typos in the revised version.

**Question 10 (1), (2): Initial case of Proof 1.**
**Re:** As depicted in Figure 1 and Eq. 7 of the main text, the $\\hat{x}_t^0$ is calculated using $\\hat{\\epsilon} _{t+1}$ from the previous step, thus to streamline the whole process. Therefore, the $\\epsilon$ is initialized at step $T+1$ and $x$ at $T$, and the initial case we need to prove is from $T$ to $T-1$, which means deriving the relationship of $\\hat{x} _{T-1}$, $\\bar{x} _{T-1}$, etc., given the relationship of  $\\|\\hat{\\epsilon} _{T} - \\epsilon _0 \\| _\infty$ obtained by imposing the PC module.

**Question 10 (3): Eq. 20.**
**Re:** Sincerely thank you for your carefulness and pointing the issue of Eq. 20. We are sorry that the sequence of Eq. 20 is mistakenly written upside-down. The correct process should be from bottom to top because $\\|\\hat{\\epsilon} _{k} - \\epsilon _0 \\| _\infty$ is actually not a conclusion but the condition that should be satisfied by applying the PC module as stated in Eq. 9, 10 of the main text. Substituting this condition into the left side of the inequality in the third row now, the inequality and the relationship of $\\|\\hat{x} _{k-1}-x _{ori}\\| _\\infty$ can be easily derived. This typographical error does not affect the correctness of the proof, and we have corrected it in the revised version.

---

### Decision · Program_Chairs · 2024-09-25

**Decision:**

Accept (poster)

**Comment:**

This manuscript studies an imperceptible adversarial perturbation generation as a non-parametric diffusion process along with solid theoretical analyses.

The reviewers' comments have been properly addressed and most reviewers are satisfied with authors' responses.
However, some experiments are somewhat tricky.
For example, the comparison results in Table 2 indicate that the proposed attack AdvAD exhibits inferior success rates than other attacks (e.g., PGD and AdvDrop) under purification-based defenses.
The use of average ASR shows bias results.
Thus, the AC disagrees with the conclusion in Lines 272~273.

On the other hand, the AC agrees with the authors' argument that ``this is the first work to theoretically derive the fundamental modeling of diffusion models for performing imperceptible adversarial attacks with inherently low perturbation strength through a novel non-parametric diffusion process, and extensive experiments have demonstrated its effectiveness.''

So, the AC tends to accept this submission!